# Sae2 controls Mre11 endo- and exonuclease activities by different mechanisms

Tomoki Tamai[1,5], Giordano Reginato [2,5], Ryusei Ojiri[1], Issei Morita[1], Alexandra Avrutis[3], Petr Cejka [2] ✉, Miki Shinohara [1,4] ✉ & Katsunori Sugimoto [3] ✉

DNA double-strand breaks (DSBs) must be repaired to ensure cell survival and genomic integrity. In yeast, the Mre11-Rad50-Xrs2 complex (MRX) collaborates with Sae2 to initiate DSB repair. Sae2 stimulates two MRX nuclease activities, endonuclease and 3′−5′ exonuclease. However, how Sae2 controls the two nuclease activities remains enigmatic. Using a combined genetic and biochemical approach, we identified a separation-of-function rad50 mutation, rad50-C47, that causes a defect in Sae2-dependent MRX 3′−5′ exonuclease activity, but not endonuclease activity. We found that both the endo- and 3′−5′ exonuclease activities are essential to release Spo11 from DNA ends, whereas only the endonuclease activity is required for hairpin removal. We also uncovered that MRX-Sae2 endonuclease introduces a cleavage at defined distances from the Spo11-blocked end with gradually decreasing efficiency. Our findings demonstrate that Sae2 stimulates the MRX endo- and exonuclease activities via Rad50 by different mechanisms, ensuring diverse actions of MRX-Sae2 nuclease at DNA ends.

Genome integrity is constantly challenged by exogenous and endogenous insults that generate various types of DNA damage[1]. DNA double-stranded breaks (DSBs) are formed as a result of exposure to exogenous agents such as ionizing radiation or chemotherapeutic agents. DSBs are also generated as byproducts after DNA replication stress or as intermediates of recombination events during meiosis and lymphocyte development. Failure to repair DSBs can result in chromosome loss, and inappropriate repair can cause chromosome rearrangements and mutagenesis. Defects in DSB repairs are associated with human disorders such as infertility, immunodeficiency, and cancer[2,3].

The Mre11 nuclease, which forms the Mre11-Rad50-Xrs2 (NBS1 in humans) complex (MRX; MRN in humans), plays multiple roles in the cellular response to DSB induction[4]. DSBs are repaired by two major pathways: non-homologous end joining (NHEJ) and homologous recombination (HR). To activate the HR pathway, DSB ends need to be converted to 3′-ssDNA-tailed DNA ends in a process termed end resection[4]. MRX collaborates with Sae2 (CtIP in humans) and initiates DNA end resection, particularly on blocked DNA ends[4]. In NHEJ, the Ku70-Ku80 complex (Ku) binds to DSB ends and then recruits other components, thereby promoting DNA end joining[5]. On the other hand, MRX removes Ku from DNA ends in an Mre11 nuclease-dependent and -independent manner[6–8]. Furthermore, MRX activates the protein kinase Tel1 (ATM in humans), which triggers checkpoint signaling through Rad53 (CHK2 in humans)[9].

End resection is a two-step process consisting of short-range and subsequent long-range resection[4]. In budding yeast, MRX-Sae2 endonuclease initiates short-range resection by introducing cleavage on the 5′ strands internal to the DSB ends. The cleavage site then serves as the initiation site for bidirectional resection: the MRX-Sae2 exonuclease

[1]Faculty of Advanced Bioscience, Graduate School of Agriculture, Kindai University, Nara City, Nara 631-8505, Japan. [2]Faculty of Biomedical Sciences, Institute for Research in Biomedicine, Università della Svizzera italiana (USI), 6500 Bellinzona, Switzerland. [3]Department of Microbiology, Biochemistry and Molecular Genetics, New Jersey Medical School, Rutgers, the State University of New Jersey, Newark, NJ 07103, USA. [4]Agricultural Technology and Innovation Research Institute, Kindai University, Nara City, Nara 631-8505, Japan. [5]These authors contributed equally: Tomoki Tamai, Giordano Reginato. ✉e-mail: petr.cejka@irb.usi.ch; mikis@nara.kindai.ac.jp; nori.sugimoto@rutgers.edu

resects DNA in the 3′–5′ direction toward the DNA end, whereas the long-range resection machinery such as the Exo1 exonuclease expands resection in the 5′–3′ direction[4]. Sae2 stimulates two MRX nuclease activities, endonuclease and 3′–5′ exonuclease[7,8,10] and coordinates the bidirectional resection at DNA ends[4]. Therefore, the endo-to-exonuclease switch of MRX-Sae2 is proposed to be essential for processing of blocked DNA ends[4]. Structural analysis shows that the Mre11-Rad50 complex binds to blocked and unblocked DNA ends in a similar configuration but in an opposite orientation[11,12], supporting the idea that Mre11 endonuclease and 3′–5′ exonuclease operate in tandem. However, it is unknown whether Sae2 promotes the MRX endonuclease and 3′–5′ exonuclease activities by the same mechanism and whether the endo-to-exonuclease switch is required to process all types of blocked DNA ends, including protein-coupled and hairpin-capped DNA ends.

Sae2 not only activates MRX nucleases but also attenuates Tel1 signaling[13,14]. We hypothesized that the Sae2-dependent controls are mediated through Rad50. We first selected mutations in *RAD50* that confer a defect in Tel1 signaling control. Among them, we further identified a separation-of-function mutation, named *rad50-C47*, that causes a defect in MRX-Sae2 3′–5′ exonuclease activity but has no effect on endonuclease function in vitro. The *rad50-C47* mutation caused a defect in the processing of Spo11-coupled DNA ends during meiosis, but no apparent defect in the repair of hairpin-capped DNA ends. Moreover, *rad50-C47* mutants were resistant to exposure to DNA-damaging agents such as Top1-poisoning camptothecin (CPT). We also observed that MRX-Sae2 introduces endonucleolytic cleavage at regular intervals adjacent to the Spo11-coupled DNA end during the meiotic prophase. These findings reveal the molecular details of the coordinated but separate endonucleolytic and exonucleolytic actions of MRX-Sae2 on blocked DNA ends.

## Results

### Isolation of a new separation-of-function *rad50-C47* mutation that causes a more limited defect than the *rad50S* mutation

Sae2 promotes MRX endonucleolytic and 3′–5′ exonucleolytic activities[7,8,10]. How Sae2 controls two different MRX nuclease activities remains to be determined. The *rad50S* mutation, located at the N-terminus of Rad50, disrupts Rad50-Sae2 interaction[10,15]. As a result, the *rad50S* mutation causes defects in MRX-Sae2 nuclease activities but also enhances the Tel1-Rad53 signaling pathway[10,13,16]. We hypothesized that Sae2 targets Rad50 to independently regulate the two MRX nuclease activities and Tel1 signaling (Fig. 1A). We thus searched for separation-of-function *rad50* mutations that retain only one nuclease activity but eliminate the other nuclease and Tel1 signaling function.

We first screened for *rad50* mutations that enhance the Tel1-Rad53 pathway after mutagenizing the *RAD50* gene in vitro by error-prone PCR[17]. To eliminate *rad50* mutations that impair Rad50-Sae2 interaction, we focused on mutations located outside of the N-terminus of Rad50 (Fig. 1B). The ATM-homolog Tel1 acts in parallel with the ATR-homolog Mec1 to activate Rad53 checkpoint signaling[18]. Enhancement of Tel1 activation makes *mec1Δ* mutant cells more resistant to DNA damaging agents or DNA replication inhibitors (Fig. S1); for example, *rad50S* or *sae2Δ* mutation restores the viability loss and the phosphorylation of Rad53 to *mec1* mutants after genotoxic stress[13,14]. Similar to the *rad50S* mutation, two *rad50* mutations (*rad50-C47* and *rad50-C126*) increased the survival of *mec1Δ* cells after exposure to hydroxyurea (Fig. 1C). The *rad50-C47* and *rad50-C126* mutations enhanced Rad53 phosphorylation in *mec1Δ* cells after phleomycin treatment (Fig. 1D), indicating that these mutations upregulate Tel1 signaling.

The *rad50-C47* and *rad50-C126* mutations contain a single amino acid substitution, at the amino acid residue 1299-K and 1297-R, respectively, in a region highly conserved among eukaryotes (Fig. 1E).

Thus, the two *rad50* mutations are located at the far end of the C-terminal ATP binding domain, which forms the Rad50 catalytic "head" with the N-terminal ATP-binding domain (Fig. 1F). Structural analysis using UCSF Chimera[19] revealed no steric clashes associated with the *rad50-C47* or *rad50-C126* mutation, suggesting that neither mutation causes significant structural alterations in the Rad50 catalytic head. No significant changes in Rad50 protein level or Mre11-Rad50 interaction were detected in *rad50-C47* or *rad50-C126* mutants (Fig. 1G, H).

If a *rad50* mutation is specifically defective in either endonuclease or 3′–5′ exonuclease, such a mutation is expected to confer a more limited defect in DSB repair compared to the *rad50S* and *sae2Δ* mutations. The *rad50S* and *sae2Δ* mutations impair Spo11 release from the meiotic DSB end and block meiotic progression[20–23]. We first examined the effect of the *rad50-C47* or *rad50-C126* mutation on meiotic progression (Fig. 2A). Both *rad50-C47* and *rad50-C126* cells progressed slowly, similar to *rad50S* cells[20]. Neither the *rad50-C47* nor the *rad50-C126* mutation significantly affected Rad50 expression during meiosis (Fig. 2B). These results suggest that both mutations cause strong functional defects in meiotic recombination.

The *rad50S* and *sae2Δ* mutations show moderate sensitivity to DNA-damaging agents[14]. We next investigated the effect of *rad50-C47* and *rad50-C126* mutation on resistance to camptothecin and phleomycin (Fig. 2C). The *rad50-C126* mutation conferred sensitivity to camptothecin and phleomycin, similar to the *rad50S* mutation. However, cells carrying the *rad50-C47* mutation were as resistant as wild-type cells.

The MRX-Sae2 pathway initiates DNA end resection but interferes with NHEJ[24]. We investigated the effect of the *rad50-C47* or *rad50-C126* mutation on the frequency of NHEJ repair using the system described[25] (Fig. 2D). In this system, the two HO cleavage sites are flanked by the *URA3* sequence in an opposite orientation in the strain that contains no donor sequence for homologous recombination. Therefore, HO-induced DSBs can be repaired only by the NHEJ pathway in this system. The *rad50-C126* mutation increased cell survival after HO cleavage, similar to the *rad50S* and *sae2Δ* mutation[24] but the *rad50-C47* mutation did not (Figs. 2D and S2). We noted that cells carrying the *rad50-C47* mutation had slightly reduced NHEJ efficiency compared to wild-type cells. However, introduction of the *rad50-C47* mutation did not cause obvious NHEJ defects in the *sae2Δ* strain (Figs. 2D and S2). Thus, the significance of the weak NHEJ defect observed in *rad50-C47* mutants is unclear. These results indicate that the *rad50-C47* mutation causes a more limited defect in MRX-Sae2 function compared with the *rad50S* mutation.

### Effect of the *rad50-C47* mutation on endonuclease and 3′–5′ exonuclease activities of MRX-Sae2 in vitro

To gain insight into the defect caused by the *rad50-C47* mutation, we purified the Rad50-C47 protein along with other MRX-Sae2 components (Fig. 3A) and examined the effect of the *rad50-C47* mutation on the endonuclease and 3′–5′ exonuclease activities of MRX-Sae2 in vitro. Phosphorylated Sae2 (pSae2) stimulates the endonuclease activity of MRX on DNA fragments with streptavidin-blocked DNA ends in vitro[16]. MRX containing Rad50-C47 mutant protein (MRX[50-C47]) exhibited a similar endonuclease activity to MRX containing wild-type Rad50 protein (MRX[50-WT]) (Fig. 3B, C). MRX-Sae2-dependent endonucleolytic cleavage creates an entry site for the Exo1 5′-3′ exonuclease[7]. Exo1 5′-3′ exonuclease activity was able to expand the degradation of the products generated by MRX[50-WT] and MRX[50-C47] to a similar extent in the presence of pSae2 (Fig. 3D). Similar results were obtained when performing the endonuclease assay on a substrate bound by the yeast Ku (Yku70-Yku80) complex, albeit with a slight defect (Fig. S3A). The *rad50-C126* mutation caused a very severe defect in Sae2-dependent activation of MRX endonuclease on streptavidin-blocked DNA ends, similar to the *rad50S* mutation (Figs. 3B, C and S3A).

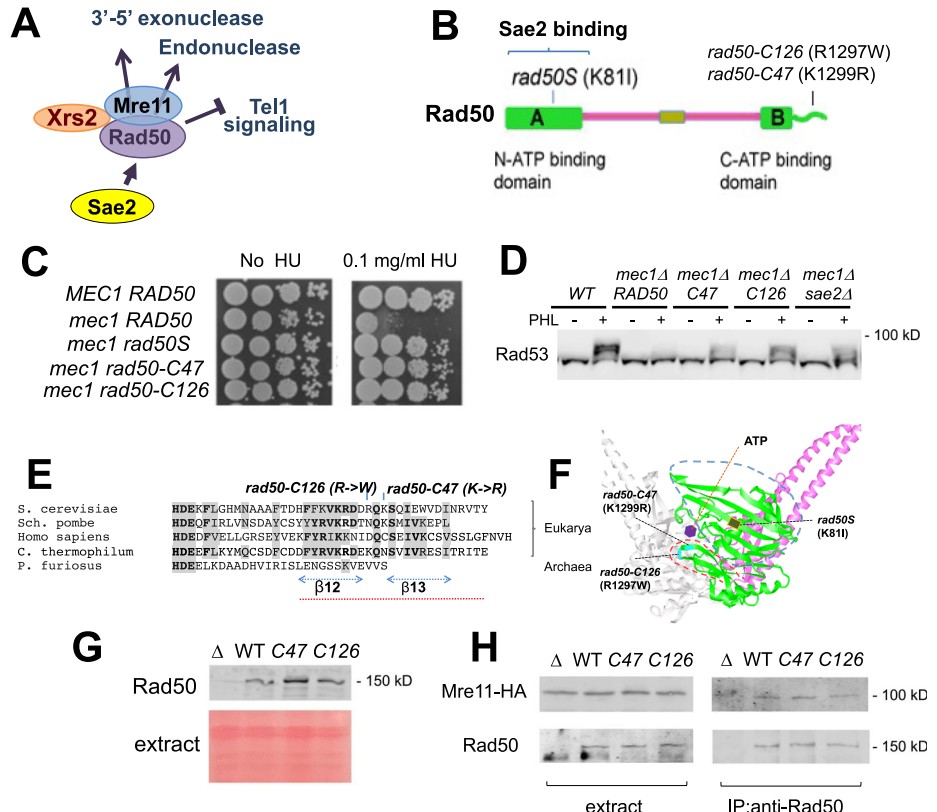

**Fig. 1 | Identification of substitution mutations in the C-terminus of Rad50 that enhance Tel1 signaling. A** Working hypothesis. Sae2 targets Rad50 and regulates two MRX nuclease activities and Tel1 signaling through distinct mechanisms. **B** Rad50 protein domains and location of *rad50* mutations. The *rad50-C47* and *rad50-C126* mutation are located at the C-terminus of Rad50. **C** Rescue of hydroxyurea sensitivity of *mec1Δ* cells by *rad50* mutations. Serial dilutions of cultures were spotted on rich medium with or without hydroxyurea (HU). **D** Enhanced Rad53 phosphorylation in *mec1Δ* cells by *rad50* mutations. Cells were treated with or without phleomycin and subjected to immunoblotting analysis with anti-Rad53 antibody. The wild-type (WT) strain, carrying both the wild-type *MEC1* and wild-type *RAD50* gene serves as a positive control. C47 or C126 denotes *rad50-C47* or *rad50-C126*, respectively. The image is a representative of two independent experiments. **E** Alignment of the C-terminus of eukaryotic and prokaryotic Rad50. Eukaryotic Rad50 proteins possess conserved C-terminal tail regions. The dotted line (in red) corresponds to the circled area in (**F**). **F** The Rad50 coiled-coil region (in pink) folds

back on itself and forms a rod-like structure, bringing the N- and C-terminal ATPase domains into the Rad50 head (in green). The diagram depicting *C. thermophilum* Rad50 was created by UCSF Chimera using the PDB data base (PDB ID: 5DAC). **G** Effect of the *rad50-C47* or *rad50-C126* mutation on Rad50 expression levels in vegetatively growing cells. Extracts were prepared from exponentially growing cells, transferred to a membrane, and stained with Ponceau S. Membranes were probed with anti-Rad50 antibodies. The strains used here carry *RAD50*, *rad50Δ*, *rad50-C47* (*C47*) or *rad50-C126* (*C126*). The image is a representative of three independent experiments. **H** Effect of the *rad50-C47* or *rad50-C126* mutation on Mre11-Rad50 interaction. Extracts were prepared from cells carrying HA-tagged *MRE11* and untagged Rad50 variants, and subjected to immunoprecipitation (IP) with anti-Rad50 antibodies. The immunoprecipitates were further analyzed by immunoblotting with anti-HA or anti-Rad50 antibodies. The strains used here carry *RAD50*, *rad50Δ*, *rad50-C47* (*C47*) or *rad50-C126* (*C126*). The image is a representative of two independent experiments.

We next examined whether the *rad50-C47* mutation affects the MRX-Sae2 exonuclease activity[10]. MRX[50-WT] and MRX[50-C47] exhibited similar basal 3′–5′ exonuclease activities (Fig. 3E, F). pSae2 stimulated the exonuclease activity of MRX[50-WT] as found previously[10]. However, pSae2 did not effectively stimulate the 3′−5′ exonuclease activity of MRX[50-C47] (Fig. 3E, F). High concentrations of pSae2 did not restore the exonuclease activity of MRX[50-C47] (Fig. 3E, F), and Rad50-C47 interacted with pSae2 similarly to wild-type Rad50 protein (Rad50-WT) in a pull-down assay (Fig. 3G), excluding the possibility that the *rad50-C47* mutation impairs MRX-Sae2 interaction. These results confirm that the *rad50-C47* mutation is the appropriate separation-of-function mutation for studying the relative importance of MRX-Sae2 endo- and exonuclease activities.

The amino-acid residue (1299-K) substituted in the *rad50-C47* mutation is located in the proximity of the C-terminal ATP binding domain (Fig. 1B). Therefore, it is possible that the *rad50-C47* mutation affects ATP binding or hydrolysis of the MRX complex. The exonuclease activity of MRX is inhibited by ATP binding[10]. We first investigated whether the *rad50-C47* mutation affects ATP-dependent

inhibition of MRX exonuclease[10]. We observed that the exonuclease activities of MRX [50-WT] and MRX[50-C47] were similarly inhibited by ATP or ATP-γ-S (Fig. 3H). Moreover, Rad50-C47 was comparable to Rad50-WT in ATP-dependent DNA binding[26] (Fig. 3I). Rif2 stimulates the ATPase activity of MRX[27]. We next examined the effect of the *rad50-C47* mutation on Rif2-mediated ATPase activation of MRX. Rif2 stimulated MRX[50-WT] and MRX[50-C47] to a similar extent, indicating the *rad50-C47* mutation does not affect the ATPase activity of the MRX complex (Fig. 3J). These results suggest that the *rad50-C47* mutation has no impact on ATP binding or hydrolysis of MRX.

To further confirm that the defect in pSae2-dependent stimulation of the MRX[50-C47] exonuclease activity was not due to impaired protein folding or MRX complex formation, we estimated the molecular weight of Rad50 variants alone or together with MX by mass photometry[28] (Fig. S3B–D). Both Rad50-WT and Rad50-C47 were dimeric by themselves and formed the Mre11-Rad50-Xrs2 complex with a 2:2:1 stoichiometry[29] (Fig. S3B–D). Together, we show that the *rad50-C47* mutation causes a strong defect in Sae2-dependent activation of MRX 3′−5′ exonuclease, but not in MRX endonuclease activation. Our results lead to a model in which Sae2 promotes the

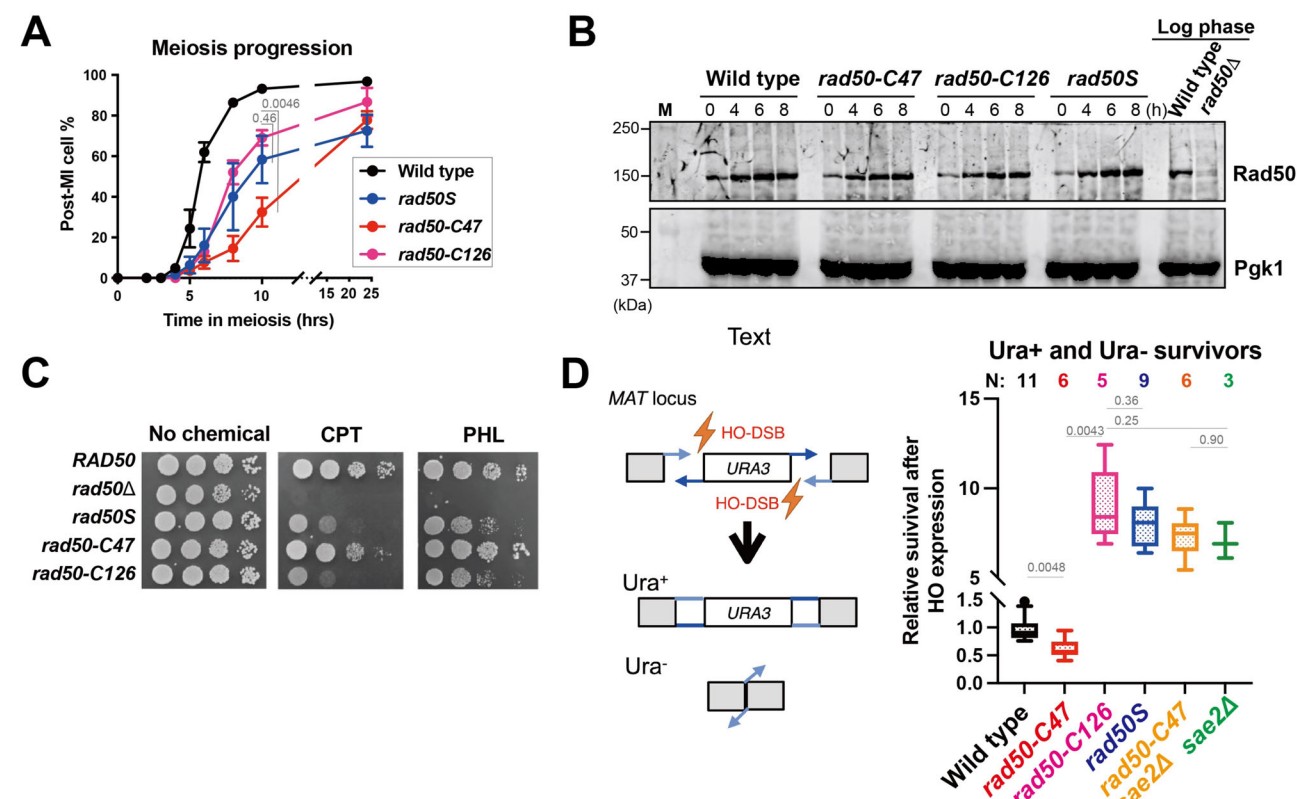

**Fig. 2 | Defects of *rad50-C47* and *rad50-C126* mutations in meiotic progression and DSB repair. A** Meiotic progression of the indicated strains. Three independent experiments were carried out. Statistical significance was determined using two-tailed Welch's *t*-test. Error bars indicate mean and standard error of the mean (SEM). **B** Effect of the *rad50-C47* or the *rad50-C126* mutation on Rad50 expression during meiosis. Cells cultured as in A were subjected to immunoblotting analysis with anti-Rad50 or anti-PGK1 antibodies. Pgk1 protein serves as a loading control. The image is a representative of two independent experiments. **C** Sensitivity of cells to DSB inducing agents. Serial dilutions of cultures were spotted on rich medium with or without 200 μM camptothecin (CPT) or 4 μg/ml phleomycin (PHL). **D** Effect

of the *rad50-C47* or the *rad50-C126* mutation on the NHEJ repair pathway. The *MAT* locus contains two HO cleavage sites in opposite orientations. Therefore, HO-induced DSBs can be specifically repaired by NHEJ. Both Ura+ and Ura− survivors were scored after HO expression and relative survival rates are shown in Tukey's box-whiskers plot. A total of four tests were carried out. In each test, two or three independent segregants from crosses were analyzed. N indicates the total number of samples analyzed in the four tests. Note that wild-type cells were used as a control in each test. Statistical significance was determined using Mann-Whitney's *U*-test using Prism10 software.

---

endonuclease and 3′-5′ exonuclease activities of Mre11 through Rad50, but by different mechanisms.

## DSB processing in *rad50-C47* mutants during meiotic recombination

The *rad50-C47* mutation causes only a slight defect in the MRX-Sae2 endonuclease in vitro. However, the *rad50-C47* cells were as defective as *rad50S* mutants in meiotic progression (Fig. 2A). We investigated the effect of the *rad50-C47* mutation on meiotic DSB processing at the well-characterized meiotic recombination hotspot *HIS4-LEU2* by comparing it with that of the *rad50S* mutation (Fig. 4A). DSB ends were processed quickly in wild-type cells. In contrast, DSB ends were accumulated in *rad50S* mutants because Spo11 remains attached at DNA ends[23] (Fig. 4B). Similar to the *rad50S* mutation, the *rad50-C47* mutation accumulated DSBs (Fig. 4B, C). However, in *rad50-C47* cells, the signal for DNA fragments containing DSBI and DSBII decreased over time (Fig. 4B, C). On the other hand, fast-migrating smeared signals increased in *rad50-C47* cells compared with *rad50S* cells after 4 hours of meiosis induction (Fig. 4B, D).

MRX-Sae2 introduces endonucleolytic cleavages near Spo11-attached DNA ends, triggering bidirectional Exo1-dependent 5′−3′ and Mre11-dependent 3′-5′ resection, ultimately removing Spo11 covalently attached to short oligonucleotides[30]. The generated ssDNA overhangs are eventually coated with the Rad51 recombinase for strand invasion[4]. We next examined the effect of the *rad50-C47*

mutation on the foci formation of Mre11 and Rad51 on meiotic chromatin spreads (Fig. 5A, B). Mre11 foci peaked at 2 hr after meiosis induction while Rad51 foci peaked at 4 hr in wild-type cells[31] (Fig. 5A, B). However, Mre11 foci remained and Rad51 foci formed very slowly in *rad50S* mutants, in agreement with the finding that the *rad50S* mutation impairs MRX-Sae2 endonuclease and hence bidirectional resection[16]. In *rad50-C47* cells, both Mre11 and Rad51 foci continued to accumulate (Fig. 5A, B). As a consequence, 70% of *rad50-C47* cells became Mre11 and Rad51 double-positive by 4 hr (Fig. 5C). Notably, introduction of the nuclease-deficient *exo1-D173A* mutation greatly attenuated the accumulation of Rad51 foci (Fig. 5D), indicating that Exo1 contributed to the majority of ssDNA production in *rad50-C47* mutants. Dmc1 is a meiosis-specific Rad51 homolog and essential for meiotic recombination[32]. Like Rad51, Dmc1 binds to ssDNA and forms discrete foci in a Rad51-dependent manner during meiosis[33]. Dmc1 foci were also detected in *rad50-C47* cells (Fig. S4). Altogether, these results suggest that Exo1-dependent 5′-3′ DNA resection occurs while Spo11 remains attached to DNA ends in *rad50-C47* cells, consistent with the in vitro data that the *rad50-C47* mutation impairs the Sae2-dependent activation of 3′−5′ exonuclease activity of MRX but does not affect the MRX endonuclease activity (Fig. 3).

We addressed if Exo1-dependent 5′-3′ ssDNA generation by itself triggers strand invasion or exchange by monitoring recombination intermediates in *rad50-C47* cells (Fig. 5E). In *rad50-C47*

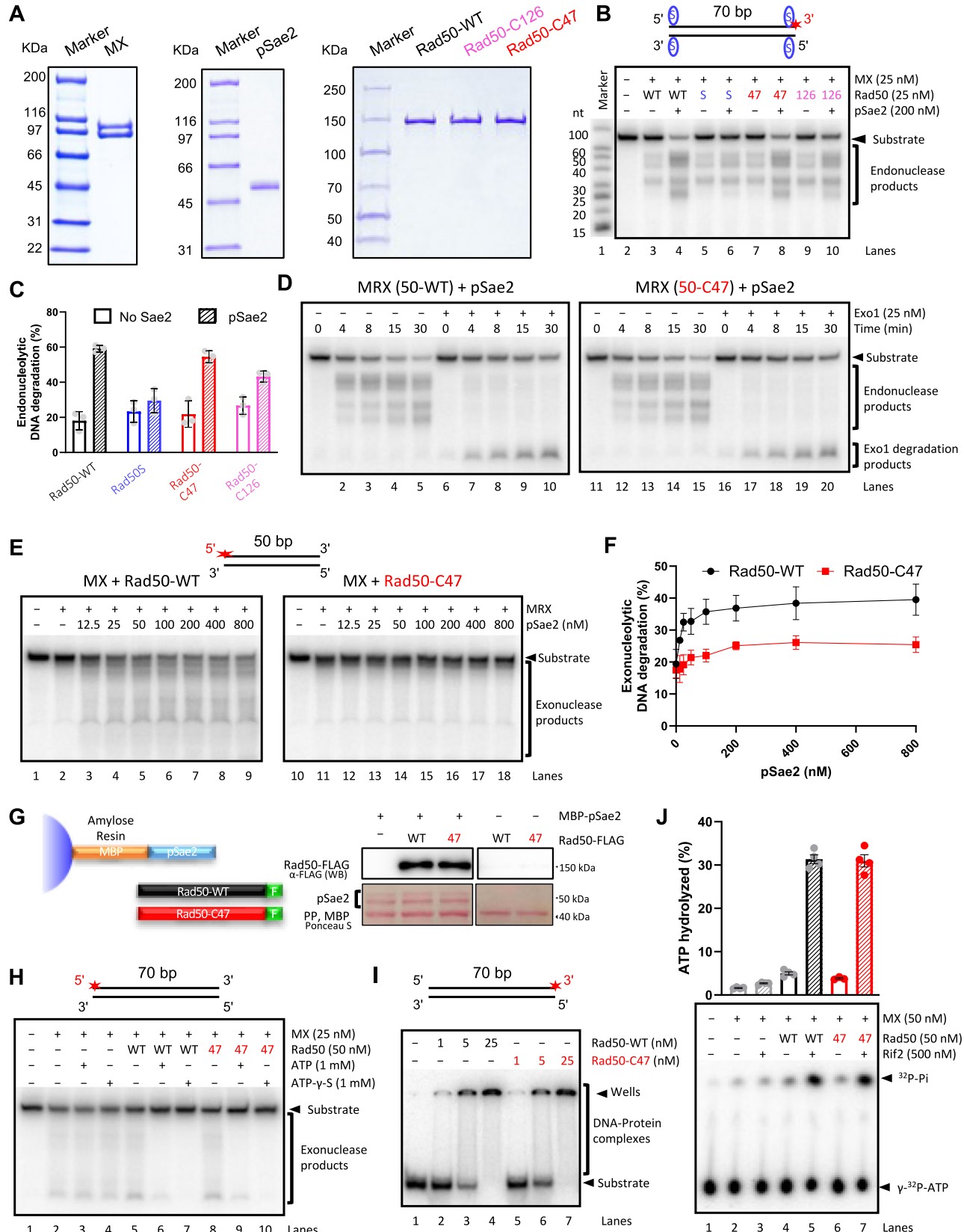

cells, recombination intermediates were barely detectable 6 hr after meiotic induction, and although the amount of intermediates increased at 8 hr, it never reached wild-type levels during the time course. Thus, Exo1-dependent 5′-3′ ssDNA generation by itself is unlikely to cause strand invasion or exchange during meiotic recombination.

## Endonucleolytic cleavage near Spo11-blocked DNA ends in *rad50-C47* mutants

MRX and Sae2 initiate short-range DNA end resection, but it is not fully understood how MRX-Sae2 initiates endonucleolytic cleavage near blocked DNA ends[4]. Our above results suggest that Spo11 remains attached to their 3′-ends after the endonucleolytic incisions by MRX-

**Fig. 3 | Effect of the *rad50-C47* mutation on the MRX-Sae2 endonuclease and 3'-5' exonuclease activity. A** Coomassie blue-stained gel of Mre11 and Xrs2 (MX), phosphorylated Sae2 (pSae2), and Rad50 variants. **B** Processing of a 70 bp streptavidin-blocked DNA substrate by MRX-Sae2 with Rad50 variants (WT: Rad50-WT, S: Rad50S, 47: Rad50-C47, 126: Rad50-C126). A cartoon of the substrate is shown. The blue oval and the red asterisk indicate the position of streptavidin and [32]P-labelling, respectively. The panel show a representative of three independent experiments. **C** Quantitation of experiments such as in (**B**). Error bars represent SEM; *n* = 3. **D** Processing of streptavidin-blocked DNA substrate by MRX-Sae2 with Rad50 variants in the absence (left) or presence of Exo1 (right). Panels show a representative of two independent experiments. **E** DNA degradation by MRX-pSae2 3'-5' exonuclease with Rad50 variants in the presence of various concentrations of pSae2. Panels show a representative of three independent experiments. **F** Quantitation of experiments such as in (**E**). Error bars represent SEM; *n* = 3.

**G** Interaction assay between Rad50-FLAG variants and pSae2. Left: a schematic of the assay. MBP-pSae2 (bait) was immobilized on amylose resin and incubated with Rad50-FLAG variants (prey). Right: a representative of two independent experiments. Ponceau S staining was used to ensure equal amount of pSae2, while Rad50-FLAG variants were visualized by Western Blotting (WB) with anti-FLAG antibody. After elution, PreScission Protease (PP) was added to the sample in order to cleave the MBP from MBP-pSae2. The left and right part of the figure are from the same exposure. **H** Exonuclease degradation of a 70 bp dsDNA by MX and Rad50 variants in the absence or presence of ATP or ATP-γ-S. A cartoon of the substrate is shown. The panel shows a representative of two independent experiments. **I** DNA binding by Rad50 variants. A cartoon of the substrate is shown. The panel shows a representative of two independent experiments. **J** ATPase activity of MX and Rad50 variants in the absence or presence of Rif2. Top: quantitation; error bars represent SEM; *n* = 4. Bottom: a representative of four independent experiments.

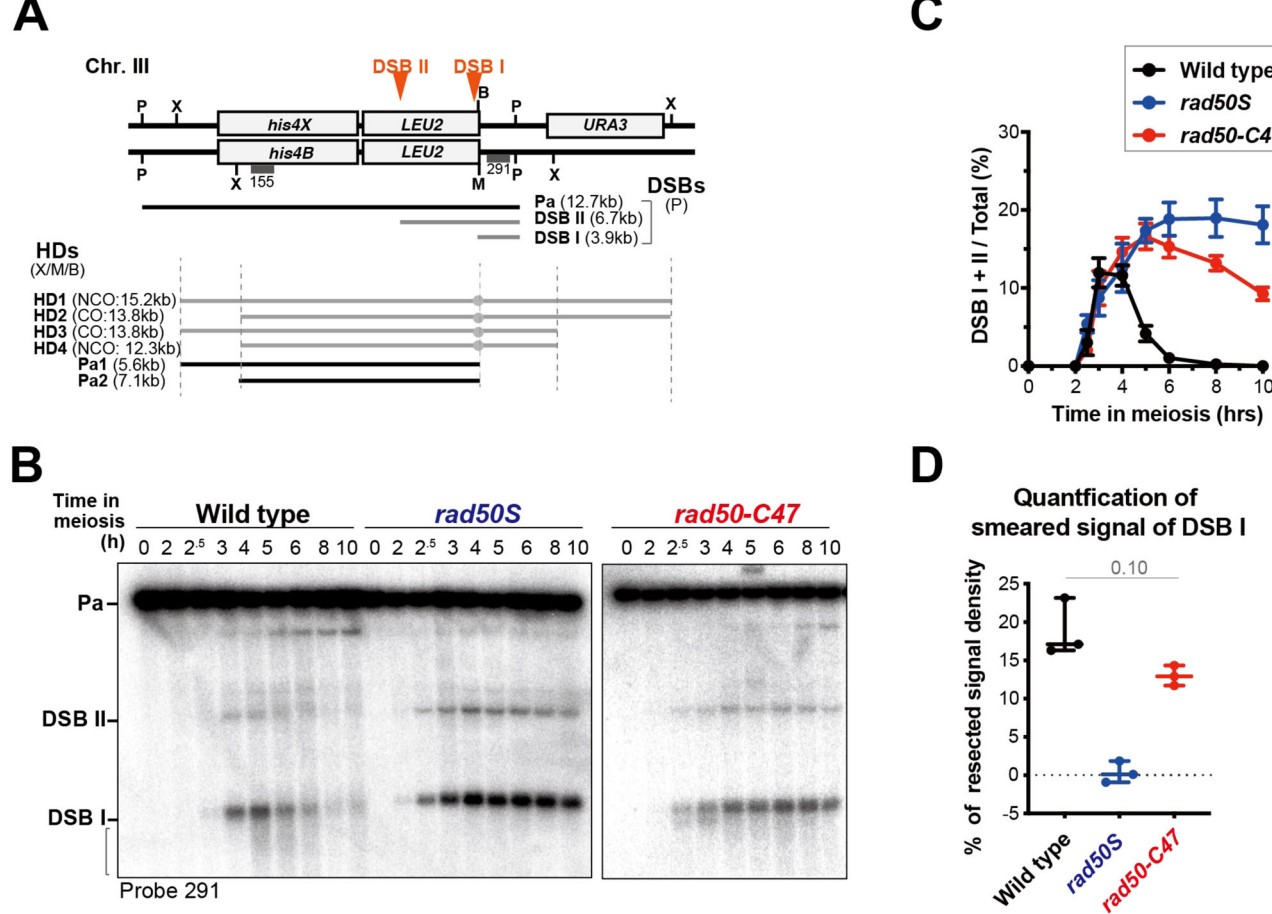

**Fig. 4 | Meiotic DSB induction and processing in *rad50-C47* mutant cells. A** Schematic of the *HIS4-LEU2* meiotic recombination hotspot on chromosome III. Diagnostic P: *Pst*I, X: *Xho*I, B: *Bam*HI, and M: *Mlu*I restriction sites are shown. The size of meiotic DSBs (DSB I, DSB II), parental (Pa), and heteroduplexes (HDs) that are associated with COs (HD2 and HD3) or NCOs (HD1 and HD4), Parental (Pa1 and Pa2), and the positions of probes 291 and 155 are shown. **B** Effect of the *rad50-C47* mutation on meiotic DSB induction. DSB induction during meiosis was analyzed by Southern blotting with probe 291. The bracket indicates non-discrete signals migrating faster than the DSBI fragment. Blot images are a representative of five independent experiments. **C** Quantification of DSB I and DSB II signals. Error bars indicate mean and SEM (*n* = 5). Statistical significance was determined using Mann-Whitney's *U*-test. The number indicates the actual *p*-values using Prism 10 software. **D** Quantification of non-discrete signals migrating faster than DSB I 4 h after meiotic induction. Error bars show maximum to minimum and median values, and all raw data points from three independent experiments are shown as closed circles. Statistical significance was determined using two-tailed Welch's *t*-test.

Sae2 in *rad50-C47* cells. To understand how MRX-Sae2 introduces endonucleolytic cuts, we monitored the formation of the Spo11-oligonucleotide complex (Spo11-oligo) and determined the oligonucleotide length of Spo11-oligos in *rad50-C47* mutants (Fig. 6A).

We first examined the effect of the *rad50-C47* mutation on the Spo11-oligo formation. Spo11 proteins were immunoprecipitated and

the associated oligonucleotides were end-labeled. The labeled Spo11-oligos were then separated by SDS-PAGE (Fig. 6B). Two bands specific for immunoprecipitated Spo11-oligos were detected in wild-type cells whereas no specific bands were observed in *rad50S* mutants, in agreement with the previous studies[23]. However, a much larger, but broader, band was detected in *rad50-C47* mutants. Spo11-oligo

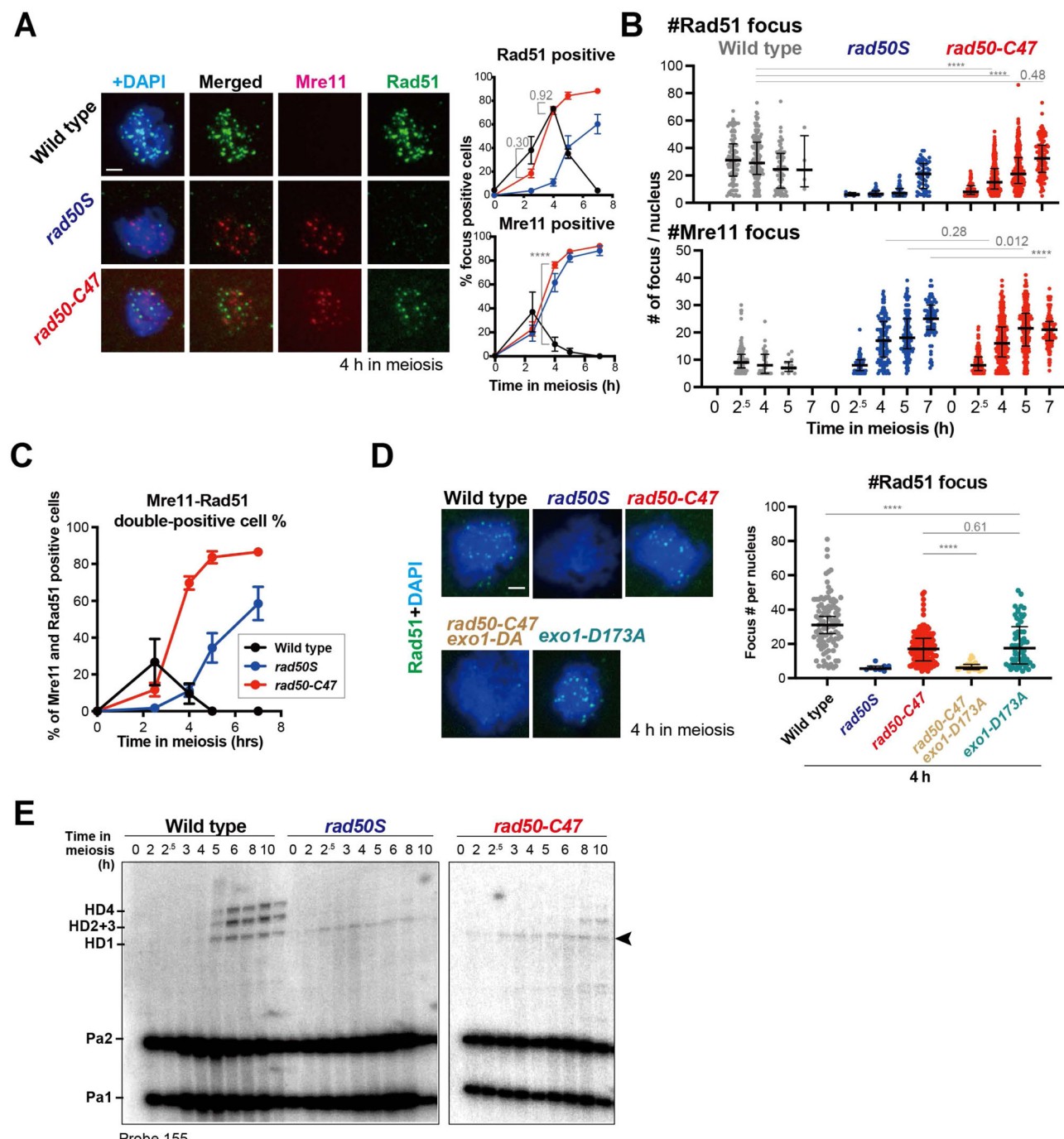

**Fig. 5 | Rad51 loading and strand invasion in *rad50-C47* mutant cells during meiotic recombination. A** Effect of the *rad50-C47* mutation on Rad51 and Mre11 foci formation during meiosis. Immunostaining images of Rad51 (green), Mre11 (red), and DAPI on meiotic nuclear spreads 4 hr after meiotic induction (Left). The percentage of foci-positive cells containing five or more foci was plotted (Right). The scale bar shows 2 μm. Error bars indicate mean and SEM from three independent experiments. At least 100 nuclear spreads were examined. Statistical significance was determined using two-tailed Welch's *t*-test. The numbers indicate the actual *p*-values using Prism 10 software. **B** Effect of the *rad50-C47* mutation on Rad51 and Mre11 foci number during meiosis. Cells were processed as in (**A**). The number of foci in each nuclear spread containing five or more foci was plotted. Error bars show median and interquartile range from three independent experiments. Statistical significance was determined using Mann-Whitney's *U*-test; ****$p < 0.0001$, otherwise, numbers indicate actual *p*-values using Prism 10 software. **C** Kinetics of Mre11 and Rad51 double-positive cells during meiosis. Cells were processed as in (**A**). The percentage of foci-positive cells containing five or more foci of both Mre11 and Rad51 was plotted. Error bars indicate mean and SEM from three independent experiments. **D** Effect of the *exo1-D173A* (*exo1-DA*) mutation on Rad51 (green) and DAPI staining in *rad50-C47* cells 4 hr after meiotic induction. Cells were processed as in (**A**). Immunostaining images (Left) and the number of foci in each nuclear spread containing five or more Rad51 foci (Right) was shown. Error bars show median and interquartile range from three independent experiments. The scale bar shows 2 μm. Statistical significance was determined using Mann-Whitney's *U*-test; ****$p < 0.0001$, otherwise, numbers indicate actual *p*-values using Prism 10 software. **E** Southern blot analysis of HD-associated CO and NCO intermediates formed at the *HIS4-LEU2* hotspot in *rad50-C47* cells. Signals were visualized by Southern blot analysis using probe 155 (See Fig. 4A). The arrowhead indicates non-specific signals overlapping with HD1 from *rad50S* or *rad50-C47* cells. Blot images are representatives of three independent trials.

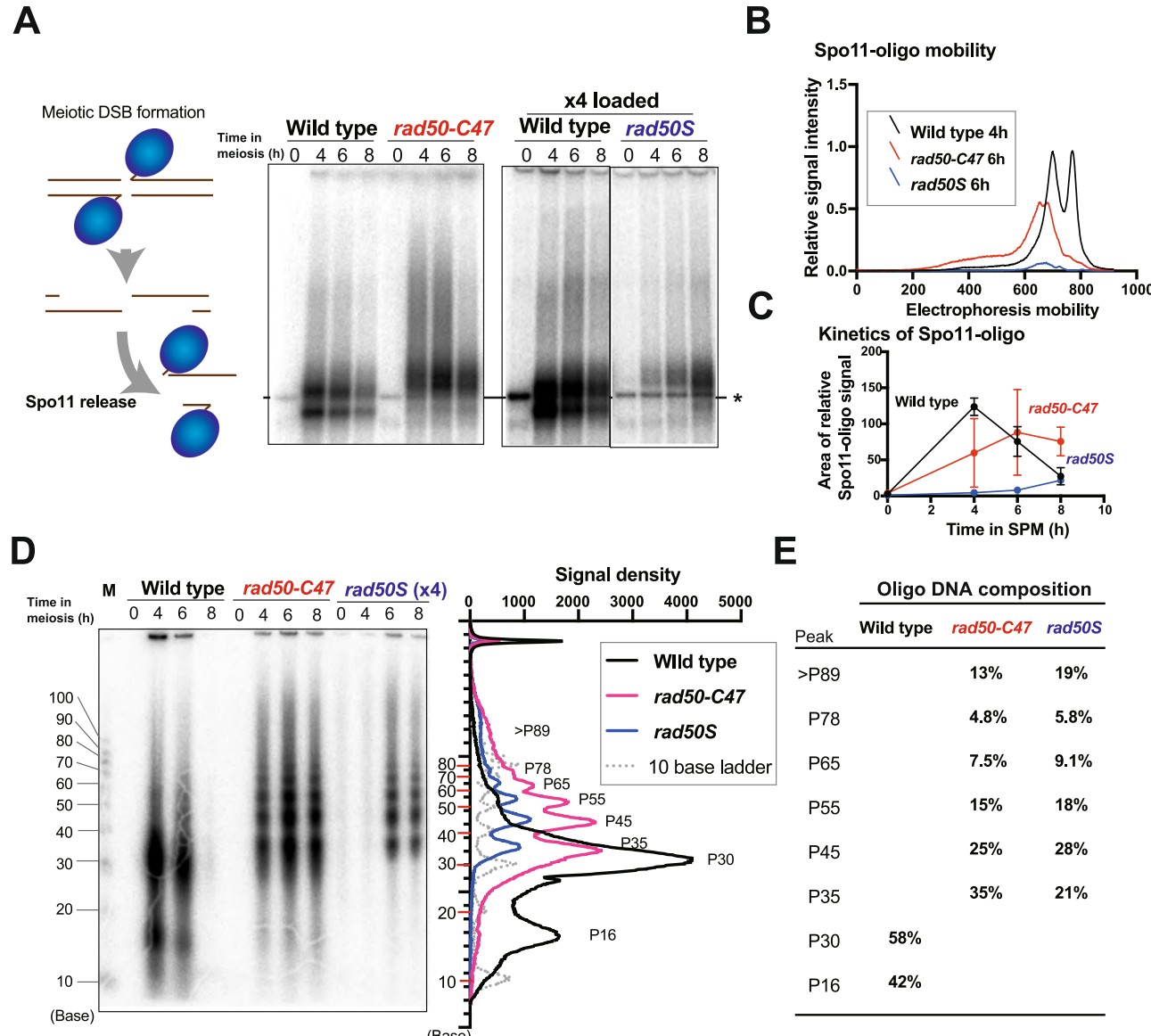

**Fig. 6 | Size of Spo11-oligos accumulated in *rad50-C47* cells. A** Accumulation of Spo11-bound oligos in *rad50-C47* cells. Spo11-bound oligos are generated by MRX-Sae2-dependent endonucleolytic cleavage (left). Spo11-bound oligos were immunoprecipitated from meiotic cells, end-labeled by TdT, and subjected to autoradiography after SDS-PAGE (right). The asterisk indicates non-specific bands derived from oligonucleotides contaminated in commercially available TdT[23]. Four times volume of immunoprecipitates were used to detect Spo11-oligos in *rad50S* cells. A representative of three independent experiments is shown. **B** Electrophoretic mobility of Spo11-oligos generated in wild-type (4 hr), *rad50S* (6 hr) and *rad50-C47* (6 hr) cells after meiotic induction. **C** Accumulation of Spo11-bound oligos during meiosis. The graph shows the kinetics of the integral (area) of

the relative value of the quantified whole lane signal intensity of Spo11-oligos. Three independent experiments were carried out. Error bars show SEM. **D** Length of oligonucleotides constituting Spo11-oligos from wild-type (4 hr), *rad50-C47* (6 hr), and *rad50S* (6 hr) cells after meiotic induction. End-labeled Spo11-oligos (**A**) were treated with proteinase K, separated on denaturing PAGE, and subjected to autoradiography (left). M denotes molecular weight marker. Signal peaks are plotted (right). Four times volume of immunoprecipitates were used to detect deproteinized Spo11-bound oligos from *rad50S* cells. A representative of three independent experiments is shown. **E** Quantification of deproteinized Spo11-oligos. Signal intensity and the length of each end-labeled DNA fragment were analyzed using ImageQuant TL software.

appearance was partially delayed in *rad50-C47* mutants compared with wild-type cells (Fig. 6B, C), suggesting that the mutation causes weak defects in endonuclease cleavage in vivo.

To further characterize MRX-Sae2 cleavage in *rad50-C47* cells, Spo11-oligos were treated by proteinase, and the resulting deproteinized oligonucleotides were fractionated on denaturing gel electrophoresis (Fig. 6D). Two peaks (P16 and P30; 16 nt and 30 nt in length, respectively) were observed in wild-type cells as found previously[30]. Neither of these peaks were detected in *rad50-C47* cells, indicating that the emergence of the two oligo species in wild-type cells depends on the MRX-Sae2 3′−5′ exonuclease action.

A ladder of bands, which results from double cuts of co-localized Spo11 (Spo11-DC), was detected in *rad50S* cells as found previously[34,35]. The -10-bp periodicity of Spo11-DC, consistent with the helical pitch of B-form DNA, suggests that Spo11 accesses only one fixed surface of DNA near the adjacent Spo11[34].

Interestingly, similar but more robust ladder bands were observed in *rad50-C47* mutant cells compared with *rad50S* mutant cells, indicating that MRX-Sae2 also accesses DNA every 10 nt near DNA-bound Spo11. However, the cleavage patterns by MRX-Sae2 and Spo11 are slightly different, consistent with the different sizes and structures of MRX-Sae2 and Spo11. In *rad50-C47* mutant cells, the shortest band

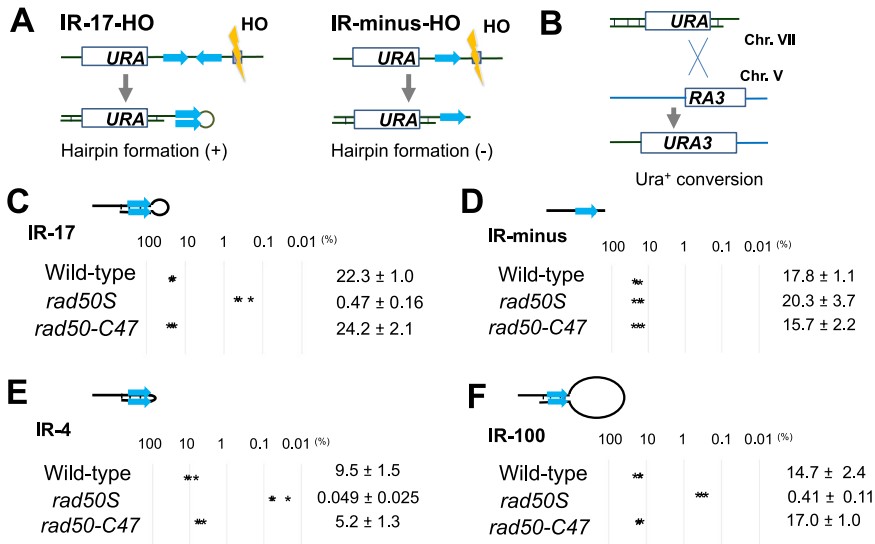

**Fig. 7 | Effect of the *rad50-C47* mutation on removal of hairpin structures from DNA ends. A** Hairpin formation from the IR-HO-17 construct after HO-induced DSB generation (**B**) Generation of a functional *URA3* gene by homologous recombination between an IR cassette on chromosome VII and a donor on chromosome V after DSB induction. **C** *URA3* recombination in cells containing the IR-17 cassette. Three independent experiments were performed and each value was plotted (asterisk). Numbers indicate the mean and standard deviation. **D** *URA3* recombination in cells containing the IR-minus cassette. Three independent experiments were performed and each value was plotted (asterisk). Numbers indicate the mean and standard deviation. **E** *URA3* recombination in cells containing the IR-4 cassette. Three independent experiments were performed and each value was plotted (asterisk). Numbers indicate the mean and standard deviation. **F** *URA3* recombination in cells containing the IR-100 cassette. Three independent experiments were performed and each value was plotted (asterisk). Numbers indicate the mean and standard deviation.

(P35) appeared the most intense and longer bands decreased with increasing size, whereas in *rad50S* mutant cells the second shortest band (P45) produced the strongest signal (Figs. 6E and S5). Thus, MRX-Sae2 seems to act on closer access points compared with Spo11. In *rad50-C47* mutants, 85% of Spo11 oligonucleotides were shorter than 90 nt (Fig. 6E), suggesting that most endonuclease cleavage occurs within 90 bp of the Spo11-attached DNA end. Notably, the pattern of the migration of the oligonucleotides in *rad50-C47* mutants did not significantly change during the time-course (Figs. 6D and S5). This observation is consistent with the finding that the *rad50-C47* mutation causes a defect in the 3′–5′ exonuclease activity, and indicates that after the initial cleavage, MRX-Sae2 endonuclease does not introduce further cleavages proximal to the Spo11-blocked ends. Thus, MRX-Sae2 endonuclease appears to move stepwise in the 5′-3′ direction from the Spo11-attached DNA end, introducing a nick and subsequently converting into a 3′–5′ exonuclease. Together, these studies using meiotic cells provide the first in vivo evidence that coordinated MRX-Sae2 endonuclease and 3′–5′ exonuclease activities are required for the processing of blocked DNA double-strand ends[4].

**Degradation of hairpin structures in *rad50-C47* mutants**
MRX-Sae2 plays a key role in the removal of hairpin structures arising from short inverted-repeats (IRs) after DNA end processing[36,37]. We further addressed whether the combination of MRX-Sae2 endonuclease and 3′–5′ exonuclease is also critical for the removal of hairpin structures.

To generate hairpin structures, we adopted an experimental system using the sequence-specific HO endonuclease[36]. In this system, inverted repeats are converted to hairpin structures after HO introduces a DSB nearby (Figs. 7A and S6). Hairpin formation depends on single-strand annealing which requires approximately 30 bp of homology for its maximum activity[38].

We first set up a system which generates hairpins with a small 17-nt loop. We constructed a cassette (IR-17-HO) consisting of the N-terminal half of the *URA3* gene, an HO cutting site and two 32 bp long sequences separated by a 17 bp spacer (Fig. 7A). The cassette which contains no IR

sequence (IR-minus-HO) was used as a control (Fig. 7A). We introduced the cassettes into the *ADH4* locus on chromosome VII in the strain containing the C-terminal half of *URA3* on chromosome V[39] (Fig. 7B). The HO endonuclease was expressed from the inducible *GAL1* promoter[40]. If HO-induced DSBs are repaired by homologous recombination between the N-terminal half and the C-terminal half of *URA3*, cells are converted from Ura⁻ to Ura⁺. Therefore, successful hairpin processing can be assessed by Ura⁺ conversion.

We examined the effect of the *rad50S* mutation on Ura⁺ conversion after HO expression (Fig. 7C, D). 20% of IR-17 and IR-minus wild-type cells generated Ura⁺ colonies after HO induction (Fig. 7C, D). While 15% of IR-minus *rad50S* mutants produced Ura⁺ cells, only 1% of IR-17 *rad50S* mutants produced Ura⁺ cells (Fig. 7C, D). These results are consistent with the current view that the MRX-Sae2 pathway plays a key role in degrading hairpin structures formed from IRs near DSBs[41] but is dispensable for uncapped DNA end processing. Interestingly, the *rad50-C47* mutation had no impact on the IR-17 hairpin resolution (Fig. 7C). Thus, hairpins with a 17 nt loop can be processed by the MRX-Sae2 endonuclease activity, independently of the subsequent 3′–5′ exonuclease activity.

The loop sizes of hairpins formed in vivo vary from very short (3-4 nt) to long (over 100 nt)[37]. We next examined whether the hairpin loop size influences MRX-Sae2 endonuclease and 3′–5′ exonuclease actions. The size of the hairpin loop can be altered by inserting spacer sequences of different sizes between IRs. We thus generated the IR repeats with a 4 or 100 bp spacer, which can be converted to a 4 nt or 100 nt loop after DSB induction, respectively (Fig. 7E, F). We next compared the effect of the *rad50-C47* and *rad50S* mutation on degradation of hairpins with a 4 or 100 nt loop by monitoring Ura⁺ conversion. Processing of DNA ends capped with a 100 nt loop was examined in the *rad1Δ* background, because MRX-Sae2 and the Rad1-Rad10 complex act redundantly to resolve hairpin structures with a large hairpin loop[36]. The *rad50S* mutation caused defects in degradation of both 4 and 100 nt loop hairpins (Fig. 7E, F). However, no significant defects were observed in *rad50-C47* mutants, irrespective of loop size. These results show that hairpin-capped DNA ends are

essentially processed by the MRX-Sae2 endonuclease, unlike Spo11-attached DNA ends, and further support the idea that Sae2 stimulates MRX endonuclease and 3′–5′ exonuclease activities through different mechanisms.

## Discussion

In this study we have isolated the *rad50-C47* mutation that confers a defect in Sae2-stimulated MRX 3′–5′ exonuclease, but not in endonuclease. We then examined the effects of this mutation on the processing of different types of obstacles at DNA ends. Several lines of evidence led to a model in which MRX-Sae2 resolves blocked DNA ends using endonuclease and 3′–5′ exonuclease in a coordinated and sequential manner. Supporting the model, Spo11 protein remained attached to DNA ends in *rad50-C47* cells during meiosis (Fig. 6). Interestingly, endonuclease cleavage occurred every 10 nt near Spo11-bound DNA ends. Therefore, MRX-Sae2 is likely to introduce ssDNA breaks at regular intervals near Spo11-bound DNA ends. Unlike in the removal of Spo11 blocks, the *rad50-C47* mutation causes no significant defects in the removal of hairpin structures at DNA ends (Fig. 7). Moreover, the *rad50-C47* mutation confers only a minor defect in the repair of Top1-trapped DNA lesions. Our studies lead to a new model in which Sae2 controls Mre11 endonuclease and 3′–5′ exonucleases via Rad50 by different mechanisms and show that the endonuclease-to-3′–5′ exonuclease switch is not required to repair all types of blocked DNA ends.

### The *rad50-C47* mutation confers a defect in MRX-Sae2 3′–5′ exonuclease, but not in endonuclease

The *rad50-C47* mutation causes defects in Sae2-stimulated MRX 3′–5′ exonuclease, but not in endonuclease activities in vitro (Fig. 3). This observation is fully consistent with in vivo data from meiotic recombination and hairpin removal analyses (Figs. 4 to 7). Structural data have shown that *E. coli* Mre11-Rad50 binds uncapped, small-hairpin-capped and protein-blocked DNA ends in a similar configuration although it binds to protein-blocked DNA ends in an opposite orientation[11,12]. Therefore, MRX-Sae2 poses similar configurations to recognize substrates but may undergo different conformational changes to exhibit endonuclease and exonuclease activity. No significant structural changes were predicted with the *rad50-C47* or *rad50-C126* mutation. However, defects in structural changes may become more apparent during endo- or 3′–5′ exonucleolytic cleavage action.

The *rad50-C47* mutation is located in the C-terminal region near the ATP-binding site but does not affect ATP binding or hydrolysis (Fig. 3H–J). The *rad50-C126* mutation, defective in both endo- and 3′–5′ exonuclease activities, is also located close to the *rad50-C47* mutation within the C-terminal region (Fig. 1B). Thus, the C-terminus of Rad50 appears to mediate Sae2-dependent MRX endo- and exonuclease activation, although the exact mechanism remains to be elucidated.

### Requirement of both MRX-Sae2 endonuclease and exonuclease for Spo11 removal from DNA ends

Spo11-oligos are accumulated in *rad50-C47* mutants during meiosis (Fig. 6), consistent with the biochemical finding that the *rad50-C47* mutation has no apparent effect on Sae2-stimulated MRX endonuclease but impairs Sae2-dependent 3′-5′ exonuclease activation. Our studies demonstrate that MRX-Sae2 accesses DNA and introduces cleavages most frequently next to the Spo11 block, but it also cleaves DNA every 10 nt from the block with progressively decreasing efficiency. It remains to be determined how MRX-Sae2 engages endonucleolytic cleavage sites after recognizing a blocked DNA end[11,42]. There are two envisaged models. One model is that activated MRX-Sae2 directly accesses the cleavage sites from the Spo11-attached DNA end (direct access model) (Fig. S7). In this model, the accessibility to each cleavage site decreases with distance from the site of Spo11-DNA. The

second model is that activated MRX-Sae2 sequentially recognizes cleavage sites as it moves along the DNA strand from the Spo11-bound DNA end in the 5′-3′ direction (sequential access model) (Fig. S7). In this model, MRX-Sae2 introduces cleavages with constant efficiency (35%) at each access point. However, it remains possible that MRX-Sae2 endonuclease activity is attenuated if it is located distantly from the Spo11-bound DNA end. It is unknown whether robust stimulation of the MRX-Sae2 endonuclease depends on direct contact with DNA end-bound obstacles.

As MRX-Sae2 introduces cleavages every 10 nt near Spo11-attached DNA ends, Spo11 double cuts occur every 10 nt[34]. Therefore, DNA-binding Spo11 and its interacting proteins might govern the DNA accessibility of other protein molecules such as MRX-Sae2 and Spo11. Similar to the *rad50-C47* mutation, the *mre11-H59S* mutation impairs Mre11 exonuclease activity[30]. However, the *mre11-H59S* mutation directly affected Mre11 exonuclease activity independently of Sae2 function[30]. Moreover, the *mre11-H59S* mutation appears to cause a milder defect in 3′–5′ exonuclease than the *rad50-C47* mutation; *rad50-C47* mutants, like *rad50S* mutants, were defective in meiotic DSB processing (Fig. 4B, C), whereas *mre11-H59S* mutants were not[30]. Indeed, the *mre11-H59S* mutation accumulated 16 and 30 nt oligonucleotides (P16 and P30)[30], which appear to be generated by MRX-Sae2 3′–5′ exonuclease activity (Fig. 6D). Interestingly, the *mre11-H59S* and *rad50-C47* mutations have different effects on site selection for endonucleolytic cleavage; *mre11-H59S* mutants accumulated long oligonucleotides ranging from 80 to 300 nt[30], whereas *rad50-C47* mutants mainly accumulated oligonucleotides shorter than 90 nt. Thus, factors other than DNA accessibility may alter MRX-Sae2 endonuclease cleavage at Spo11-bound DNA ends.

Our results support the idea that MRX-Sae2 endonuclease introduces cleavages without pronounced sequence preference near Spo11-attached DNA ends. Spo11 binds to the nucleosome-free regions for DSB induction[43]. However, DNA breaks can be also generated within nucleosome-rich regions. Recent studies show that MRX-Sae2 cleavage is affected by chromatin structures on chromosomes[44].

While ssDNA was generated, meiotic recombination was nearly blocked in *rad50-C47* mutants. One explanation is that Spo11 retained at the DNA end or a short unresected double-stranded DNA region near Spo11 prevents strand invasion or exchange in *rad50-C47* mutants during meiosis. This is consistent with the observation that hairpin formation interferes with homologous recombination[41] (Fig. 7). Both Rad51 and Dmc1 formed foci in *rad50-C47* mutants (Figs. 4 and S4). However, Dmc1 and Rad51 appear to be localized to different regions of ssDNA filaments[45,46]. It remains possible that uncoupled ssDNA binding of Rad51 and Dmc1 affects strand invasion or exchange in *rad50-C47* mutants.

### Requirement of MRX-Sae2 endonuclease activity for hairpin removal

Our studies demonstrated that MRX-Sae2 recognizes and processes hairpin-capped DNA ends differently than protein-blocked DNA ends. MRX-Sae2 collaborates with the ssDNA-binding protein RPA to prevent foldback formation at DSBs[47]. Binding of RPA to ssDNA overhangs or hairpin structures stimulates MRX-Sae2 endonucleolytic cleavage 25–40 nt away from the DNA end[7,8]. RPA would bind more efficiently to larger single-stranded hairpin loops than smaller ones. Thus, it has been predicted that MRX-Sae2 endonuclease and 3′–5′ exonuclease sequentially process large hairpin-capped DNA ends like protein-blocked DNA ends[8,48]. However, our studies of *the rad50-C47* mutation show that the MRX-Sae2 endonuclease activity is required for the removal of hairpins of various sizes from 4 to 100 nt, but the 3′–5′ exonuclease activity is not (Fig. 7). To degrade hairpin structures, the MRX-Sae2 endonuclease activity may be coupled with DNA melting[49] or may introduce incisions immediately next to hairpin loop structures[50]. Binding of RPA to single-stranded hairpin loops might

promote DNA melting or ensure proximal incision by MRX-Sae2 endonuclease. Alternatively, other proteins might substitute for MRX-Sae2 3′–5′ exonuclease in this context.

Cells harboring the *rad50-C47* mutation had no apparent defect in the cellular response to CPT treatment that produces DSBs with Top1 covalently attached to the 3′-terminus (Fig. 2C). The MRX-Sae2 endonuclease introduces cleavage at the 5′-terminated strand even when the 3′-end is blocked[16]. At the moment, however, it is not known how MRX-Sae2 endonuclease processes DNA lesions bound by Top1 at the 3′-terminus. Cells carrying the *rad50-C47* mutation were also as resistant to phleomycin as wild-type cells (Fig. 2C). MRX-Sae2 endonuclease might play a key role in removing hairpin structures generated after exposure to DNA-damaging agents such as CPT and phleomycin.

### Link to NHEJ and Tel1 signaling activation

The *rad50S* mutation enhanced NHEJ, whereas the *rad50-C47* mutation did not (Fig. 2D). Thus, the *rad50-C47* mutation does not impair the regular HR-NHEJ pathway choice. There were no apparent differences in Ku accumulation between *rad50S* and *rad50-C47* mutants (Fig. S8). DNA end resection has been proposed to suppress the NHEJ pathway[51]. Our studies further indicate that 5′−3′ degradation alone, rather than bidirectional degradation, plays a critical role in the NHEJ inhibition. Similar to the *rad50S* mutation, the *rad50-C47* or *rad50-C126* mutation enhanced Tel1 signaling after DNA damage. Both the *rad50-C47* and *rad50-C126* mutations carry substitution mutations at the C-terminus. The current model suggests that Sae2 inhibits Rad9-Rad53 interaction near DNA damage sites, thereby attenuating Tel1 signaling[14,52]. Sae2 may regulate Rad9-Rad53 interaction through the C-terminus of Rad50.

In summary, our studies have revealed that Sae2 controls two MRX nuclease activities through Rad50 but by different mechanisms, ensuring versatile actions of MRX-Sae2 nuclease depending on the characteristics of the DNA ends. The Mre11-Rad50 pathway is a highly conserved system during evolution. It will be interesting to see how the functions of MRX-Sae2 are integrated and whether similar regulation is conserved in other organisms.

## Methods

### Plasmids and strains

The pFB-RAD50 plasmid carrying the *rad50-C47* mutation was constructed as follows. Fragments containing the N-terminus and C-terminus of the *rad50-C47* mutation were amplified by PCR with the primer pairs (KS3731 and KS3730 or KS3729 and KS3732) using the pFB-Rad50-FLAG plasmid as a template and fused to BamHI-XhoI-treated pFastBac1 (ThermoFisher) using In-Fusion® Snap Assembly (Takara). The pFB-RAD50 plasmid carrying the *rad50-C126* mutation was constructed as follows. Fragments containing the N-terminus and C-terminus of the *rad50-C126* mutation were amplified by PCR with the primer pairs (KS3731 and KS3724 or KS3723 and KS3732) using the pFB-Rad50-FLAG plasmid as a template and cloned into BamHI-XhoI-treated using In-Fusion® Snap Assembly.

The KanMX-*URA3ΔC-HO* cassettes containing various inverted-repeats (IR-minus, IR-4, IR-17 and IR-100) were constructed as follows. The *ADH1*-driven *K. lactis* URA3 (*ADH1-URA3*) gene was amplified by the primer pair (KS2181 and KS2182, KS3845, or KS3873) using pNO-URA3[39] as a template, and the resulting fragments were digested with XhoI and EcoRI, and then cloned into XhoI and EcoRI digested pTG22-HO-CA[53] to generate the IR-minus, IR-17 or IR-100 plasmid, respectively. To construct the IR-4 cassette, the *ADH1-URA3* gene was amplified by PCR with the primer pair (KS2181 and KS3872) and treated with XhoI. The XhoI-treated fragment and the annealing products of oligonucleotides KS3870 and KS3871 were cloned together into XhoI-EcoRI-digested pTG22-HO-CA[53] to generate the IR-4 plasmid. The IR plasmids were digested with NotI and SalI, and the IR cassettes were integrated into the *ADH4* locus on chromosome VII. The *HphMX*-marked ura3-ΔN cassette was introduced into the *YER186* locus on chromosome V[39,54]. The *rad1Δ* mutation was prepared by a PCR-based method[55,56] using the primer pair (KS865 and KS866).

To integrate the *rad50-C47* and *rad50-C126* mutation into the own locus, the mutation genes were amplified by PCR with the primer pair (KS3548 and X022). The resulting fragments were fused by PCR to the *K. lactis URA3* gene for integration[57]. The *URA3* marker was subsequently replaced with the *TRP1*, *KanMX* or *HphMX* marker[54,56]. The HA-tagged *MRE11* strains were prepared using the primer pair (KS1013 and KS1014) as described[56]. Each strain construction was confirmed by PCR with specific primers or by phenotype analysis. The sequences of the PCR primers used for the authentification PCR procedures will be provided upon request.

### Genetic screen

The C-terminal half of *RAD50* was amplified by error-prone PCR with the primer pair (M13 reverse and KS3511) using pRS314 carrying Rad50 as a template in the presence of 0.1 mM MnCl$_2$, and the N-terminal region was amplified by regular error-free PCR with the primer pair (M13 forward and KS3512)[17]. PCR products were introduced together with BamHI-digested YCplac33 into the *mec1Δ sml1Δ* strain (KSC2244). Colonies on uracil-drop out medium were replica-plated on medium containing 1 mg/ml of hydroxyurea (HU). Plasmids were recovered from the colonies and retested. Out of ~20,000 transformants, thirteen plasmids were found to support proliferation on medium containing HU. The *rad50-C47* and *rad50-C126* mutations were recovered four and three times, respectively. Both mutations rescued the HU sensitivity of *mec1* mutants more effectively than the remaining clones.

### Noncomplementary DSB rejoining assay

Non-homologous end-joining activities to repair HO-induced DSBs were analyzed using SLY19 derivatives[25,58]. A single colony was inoculated and cultured in raffinose for 14 hr. Aliquots were plated on glucose after serial dilutions to estimate cell number. The remaining cultures were cultured after the addition of galactose to a final concentration of 2% (w/v). After 2.5 hr incubation, cells were plated on galactose. To determine the frequency of Ura$^{+/-}$ prototrophs, cells grown on galactose were replica-plated onto uracil-dropout medium.

### Recombination assay of HO-induced DSBs

Cells were transformed with the *ADE1*-marked GAL-HO plasmid[53] and grown in adenine- and histidine-dropout medium containing 2% sucrose and 0.1% glucose overnight. The culture was then diluted 20-fold and grown in adenine-dropout medium containing 2% galactose or 2% sucrose and 0.1% glucose for 4 hr. Aliquots of the cultures were diluted and plated on uracil-dropout or non-selective medium to estimate the *URA3* recombination frequency before and after HO expression. Experiments were carried out at least three times for statistical evaluation.

### Meiotic cell progression

Meiotic time course experiments were carried out using SK1 isogenic NKY1551 derivatives[59] as described[60,61]. Cells cultured in sporulation medium (0.3% potassium acetate, 0.02% raffinose) were fixed with 70% ethanol and stained with 4′,6-diamidino-2-phenylindole (DAPI), and cells containing more than two nuclei signals were counted as post-MI cells.

### Cytological analysis

Chromosome spreads were prepared by the Lipsol method and analyzed by immunostaining with guineapig anti-Rad51 or rabbit anti-Mre11 antibodies[62,63]. AlexaFluor488 anti-guinea pig IgG (A-11073, Invitrogen) and CF568 anti-rabbit IgG (20098-1, Biotinum Inc.) were used for visualization. More than 100 nuclei were analyzed at each time

point, and nuclei containing more than 5 foci were scored as foci positive.

## Southern blot analysis of meiotic DSBs and intermediates

Genomic DNAs were digested with PstI to detect DSB induction and with XhoI, BamHI, and MluI to detect meiotic intermediates, respectively. Digested DNAs were transferred onto a nylon membrane and hybridized with radio-labeled DNA fragments as described[64]. Probes were labeled by Klenow fragment (New England Biolabs) with [α-$^{32}$P] dATP (PerkinElmer) after annealing with random hexanucleotides (New England Biolabs). $^{32}$P signals were detected and analyzed by a phosphorimager Typhoon FLA7000 using ImageQuant TL software (Cytiva).

## Detection of Spo11-oligonucleotides and deproteinized oligonucleotides

Spo11-oligos were immunoprecipitated from yeast SK1 strains containing the *SPO11-FLAG* allele as described[64,65]. Spo11-FLAG protein was immunoprecipitated using an anti-DYKDDDK tag antibody (1E6, Fujifilm-Wako) and Dynabeads Protein G (Veritas) in a buffer (2% triton X-100, 30 mM Tris-HCl pH8.0, 300 mM NaCl, 2 mM EDTA, 0.02% SDS). Immunoprecipitated Spo11-oligos were labeled at the 3' end by terminal deoxynucleotidyl transferase (TdT, Takara) and [α-$^{32}$P]dCTP (Perkin Elmer). The reaction was terminated by the addition of the SDS-PAGE loading buffer. To analyze the Spo11-oligo formation, samples were separated on SDS-PAGE. To determine the size of deproteinized oligonucleotides, samples were further treated with proteinase K at 65 °C for 60 min, extracted with phenol/chloroform solution and precipitated in ethanol in the presence of 1 mg/ml of glycogen. Oligonucleotide ladders (G4471, Promega) were labeled with α-$^{32}$P-dCTP by TdT as above. Deproteinized samples and DNA size marker were denatured in formamide dye (Fisher Scientific) and resolved in 15% TBE-Urea gel (Novex EC68855, ThermoFisher). $^{32}$P signals were detected and analyzed by a Typhoon phosphorimager (Cytiva).

## DNA substrates for nuclease assays

The oligonucleotide-based substrates (70 bp) were prepared by annealing radio-labeled PC210 oligonucleotide with a two-fold excess of the PC211 oligonucleotide. For endonuclease and DNA-binding reactions, PC210 was labeled at the 3'-end using TdT (New England Biolabs) with [α-$^{32}$P]dCTP (PerkinElmer) following the manufacturer's instructions and purified with Micro Bio-Spin P-30 Gel Columns (Bio-Rad). Exonuclease assays were performed using a 50 bp substrate obtained with X12-3 oligonucleotide labeled at the 5'-end with T4 polynucleotide kinase (New England Biolabs) and [γ-$^{32}$P]ATP (PerkinElmer), purified and annealed with a three-fold excess of X12-4C oligonucleotide. Where indicated, exonuclease assays were performed with a 70 bp substrate obtained by annealing 5'-labeled PC210 with a 2-fold excess of PC211.

## Protein expression and purification

All recombinant proteins were expressed from *Spodoptera frugiperda* (*Sf*9) cells. The yeast Mre11-Xrs2 (MX) complex was expressed using pTP391 and pTP69473 (Tanya Paull, University of Texas at Austin) expressing His-tagged Mre11 and FLAG-tagged Xrs2, respectively[66]. The complex was purified by NiNTA and anti-FLAG affinity chromatography[67]. The Rad50-FLAG variants were expressed in *Sf*9 cells using pFB-RAD50-FLAG vector (WT and variants) by anti-FLAG affinity chromatography. Phosphorylated Sae2 (pSae2) was expressed using the pFB-MBP-Sae2-His vector expressing MBP and His-tagged Sae2 in the presence of phosphatases inhibitors and purified by amylose and NiNTA affinity purification[15,27]. The MBP tag was removed using PreScission Protease before the NiNTA affinity purification step. Exo1 was expressed using the pFB-Exo1-FLAG and purified by FLAG affinity and HiTrap SP HP (Cytiva) ion exchange chromatography[67].

The Ku complex was expressed with pFB-MBP-Ku70-his and pFB-Ku80-FLAG coding for MBP- and his-tagged Ku70 and FLAG tagged Ku80, respectively[7]. The complex was purified using amylose and FLAG affinity resins. The MBP tag was removed by incubation with PreScission Protease before the FLAG purification step.

## Nuclease assays using MRX-Sae2 and Exo1

Nuclease assays (15 µl volume) were performed in a reaction buffer containing 25 mM Tris-acetate pH 7.5, 1 mM dithiothreitol, 5 mM MgCl$_2$, 1 mM MnCl$_2$, 1 mM ATP, 80 U/ml pyruvate kinase (Sigma), 1 mM phosphoenolpyruvate, 0.25 mg/ml bovine serum albumin (New England Biolabs) and 1 nM DNA substrate (in molecules). For the effect of ATP on the exonuclease activity of MX + Rad50, magnesium was omitted, manganese was used at 5 mM concentration and ATP or ATP-γ-S were added as indicated. The reactions were then mixed and incubated at 30 °C for 30 min. For the experiments with 3'-labeled 70 bp substrates, the mix was supplemented with 30 nM monovalent Streptavidin (a kind gift from M. Howarth, University of Oxford) or 10 nM Ku complex, as indicated, and incubated for 5 min at 25 °C (30 °C for the Ku complex) before the addition of the other proteins. Reactions were stopped by the addition of 1 µl of stop mixture (5% SDS and 250 mM EDTA) and 0.5 µl of 20 mg/ml proteinase K (Roche) followed by incubation for 60 min at 50 °C. For the time-course experiment with Exo1, a master mix was assembled and the proteins were added on ice, as indicated. At the indicated time, 15 µl of the reaction were collected, stopped and deproteinated as described above. Deproteinated samples were denatured by boiling at 95 °C for 5 min after addition of 16.5 µl of urea loading dye (95% formamide, 20 mM EDTA and 1 mg/ml bromophenol blue) and separated on 7 M urea-15% acrylamide gels. Gels were dried and analyzed using a Typhoon phosphor-imager (GE Healthcare). Gel quantifications were performed with the ImageJ software.

## Rad50-Sae2 interaction

Extracts from *Sf*9 insect cells expressing phosphorylated MBP-Sae2 were mixed with 50 µl of amylose resin and incubated for 1 h at 4 °C with constant agitation. After incubation, the resin was washed 5 times with 1 ml wash buffer (50 mM Tris-HCl pH 7.5, 2 mM EDTA, 80 mM NaCl, 0.2% NP40 and 1:400 v/v protease inhibitor cocktail [Sigma P8340]) and incubated for 1 h at 4 °C with 2.5 µg of purified FLAG-tagged Rad50 variants. Subsequently, the resin was washed 4 times with 1 ml of wash buffer, and proteins were eluted using wash buffer supplemented with 20 mM maltose. 2 µg of PreScission protease was added to the sample and incubated for 1 h at 4 °C. Proteins were separated by SDS-PAGE and analyzed by Ponceau S staining and immunoblotting with anti-FLAG M2 antibody (F3165, Sigma).

## ATPase assay

ATPase assays with Rad50 variants (15 µl) were carried out in a reaction buffer containing 25 mM Tris-HCl pH 7.5, 5 mM magnesium acetate, 1 mM DTT, 0.25 mg/ml bovine serum albumin (New England Biolabs), 0.15 mM ATP, 1 nM of [γ-$^{32}$P] ATP (Hartmann-Analytic) with 70 bp dsDNA as a co-factor (50 nM, in molecules ). The samples were supplemented with the indicated proteins on ice, mixed and incubated at 30 °C for 2 h. Reactions were stopped with 1.69 µl of 0.5 M EDTA and separated on thin layer chromatography plates (Merck) with 0.3 M LiCl and 0.3 M formic acid as the mobile phase. Plates were dried, exposed to storage phosphor screens (GE Healthcare) and scanned by a Typhoon phosphorimager. ImageJ software was used to quantitate ATP hydrolysis and the values were plotted using GraphPad Prism.

## Electrophoretic mobility shift assay (EMSA)

DNA binding of Rad50 variants were carried out in a binding buffer containing 25 mM Tris-acetate pH 7.5, 5 mM magnesium chloride, 1 mM dithiothreitol (DTT), 0.25 µg/µl bovine serum albumin (New

England Biolabs), 1 mM ATP-γ-S, and 70 bp dsDNA substrate (1 nM, in molecules). The indicated concentrations of Rad50 variants were incubated for 15 min at 30 °C. The reactions were supplemented with 5 µl of EMSA loading dye (50% glycerol [w/vol] and bromophenol blue) and separated on 6% polyacrylamide gels (ratio acrylamide:bisacrylamide 19:1, Bio-Rad) in TAE buffer (40 mM Tris-HCl, 20 mM acetic acid and 1 mM EDTA) on ice. The gels were dried on 17 CHR papers (Whatman), exposed to a storage phosphor screen (GE Healthcare) and scanned by a Typhoon Phosphor Imager. DNA binding was measured using ImageJ and plotted with GraphPad Prism.

## Immunoblotting analysis
Immunoblotting analyses were carried out by electronic transfer of proteins separated after SDS-PAGE onto membranes[17,61]. Mitotic cells were processed by an alkali lysis method[68]. Meiotic cells were treated with trichloroacetic acid and cell extracts were prepared using a Multi-Beads Shocker cell disrupter (YASUIKIKAI, Japan)[61]. Blots were probed with anti-Rad50 (1/2000, a gift from John Petrini, Memorial Sloan Kettering Memorial Cancer Center), anti-Pgk1 (1/5000, 22C5D8, Abcam) or anti-HA (1/2000, 16B12, BioLegend) and signals were visualized on the Odyssey Infrared Imaging system (Li-Cor).

## Other methods
DNA damage sensitivity and immunoprecipitation experiments were carried out as described[17]. The structural change of Rad50 mutant proteins was evaluated using the Rotamers tool of UCSF chimera (https://www.cgl.ucsf.edu/chimera/).

## Reporting summary
Further information on research design is available in the Nature Portfolio Reporting Summary linked to this article.

## Data availability
Raw images or raw numbers supporting the plotted data are provided in the source data files. Source data are provided with this paper.

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

## Acknowledgements

We thank John Petrini and Akira Shinohara for sending materials, Patrick Sung for discussion at the early stage, Samuel Bunting for critical reading, Tiffany Brown, Ke Li, Megumi Marumoto and Anibian Rodriguez for technical assistance. This work was supported by the Swiss National Science Foundation (SNSF) (Grants 310030_207588 and 310030_205199) (P.C.), the European Research Council (ERC) (Grant 101018257) (P.C.), Takeda Science Foundation (M.S.), Grant-in-Aid for Scientific Research from the Japan Society for the Promotion of Science (JSPS) (19K22402) (M.S.), NIH R01GM120730 (K.S.), Rutgers Global Grant (K.S.), Rutgers Busch Grant (K.S.) and RBHS Bridge Grant (K.S.). Part of the research in the Shinohara laboratory was carried out under the Cooperative Research Program as the Visiting Fellow (VFCR-21-03) and Collaborative Research Program (CR-23-03) of Institute for Protein Research, Osaka University.

## Author contributions

T.T., I.M., R.O. and M.S. were involved in the NHEJ assay and meiotic cell analysis. A.A. and K.S. carried out genetic screening, genetic and biochemical assays using mitotic cells as well as structural analysis of Rad50 mutant proteins. G.R. and P.C. purified proteins and performed in vitro assays. G.R., P.C., M.S. and K.S. participated in project planning and writing. K.S. drafted and completed the manuscript.

## Competing interests

The authors declare no competing interests.
