## [Peer Review File · Nature Communications]

Sae2 controls Mre11 endo- and exonuclease activities by different mechanismsREVIEWER COMMENTS

Reviewer #1 (Remarks to the Author):

The manuscript entitled Sae2 controls Mre11 endo- and exonuclease activities by different mechanisms authored by Tomoki Tamai et al. The authors provide data that convincingly support the title of the manuscript regarding different mechanism of control by Sae2 of the endo and exonuclease activities of Mre11. The authors identified a separation of function rad50 mutant that is specifically defective in the RMX exonuclease activity. This allowed the author to show that both exo and endonuclease activity is needed to remove Spo11 from the ends of DNA. Contrastingly, the ability to remove hairpins by RMX relies on the endonuclease activity of Mre11. The author show that Sae2 promotes RMX cleavage at distinct distances from the Spo11-DNA end.

The results presented in this manuscript are significant and provide valuable insight to the regulation of RMX by Sae2. The work answers some unresolved questions in how RMX removed Spo11 from the ends of DNA. The work will be of great interest to the DNA repair field, NHEJ, and meiosis. The work supports the claims of the authors. Some of the text was a little unclear and could be revised to improve clarity for a broad audience. The authors interpretation of the data is logical. The methodology is sound. Some of the quantitation of the images is not in agreement with the images that are presented. This is likely a minor issue as the experiments were repeated multiple times and could be resolved by using images that are more representative of the data in the graphs. The quality of the work exceeds those expected in our field. The purified proteins and the in vitro digestion gels are beautiful. The authors have a number of methods that reference previous work. The description of the methods would allow someone in their field to replicate the experiments. Although my enthusiasm for this manuscript is relatively high, there are some concerns and some inconsistencies between the text and the data. I encourage the authors to address the concerns listed below as they will help with clarity of work.

Comments:

1. The authors performed structural analysis using UCSF Chimera and concluded that neither C47 and C126 mutations cause significant alterations. There was no mention of what structural analysis was performed. For instance, AlphaFold may not model the nuanced changes that could occur with a R to W or a K to R substitution. There are plenty of examples in the literature that involve a single amino acid change that results in changes in the structure of a protein that results in unexpected changes in the proteins behavior. While unlikely as the authors conclude, have the authors considered that the mutations may alter the affinity of Rad50 for Mre11-Xrs2 complex or the interaction of RMX with Sae2? Do the authors have results demonstrating that these mutants of RAD50 have the same affinity for M-X and in the complex of RMX interacting with Sae2. Likewise, is it known whether the C47 and C126 mutations alter ATP affinity, ATP hydrolysis rates or DNA binding activity of RAD50. Given the proximity of the C47 and C126 mutations to the ATP binding pocket per Figure 1, it is possible that perturbation of any of these activities could provide clues to the mechanism by which Rad50 influences Mre11 activities with Sae2 present.
2. Page 7 and Figure 3. The authors clearly show there are two different mechanisms by which Sae2 promotes endonuclease and exonuclease activities of Mre11 via Rad50. The purified proteins are exceptionally pure and the nuclease digestion gels are beautiful.
3. In Figure 4B,C,D, the authors indicate that the signal for DNA fragments containing DSBI and DSBI decrease over time for the rad50-C47'. However, in the gel in Figure 4B, the DSB I and II is increasing over time for the rad50-C47 until a slight decrease at 10 hours. The data plotted in C for rad50-C47 do not reflect the band intensity in panel B. The authors may have selected the wrong representative gel to use in panel B?
4. Figure 4D and B, the authors state that the faster migrating smears increased with rad50-C47. Please clarify what they were compared to and what they increased from? According to the gel in panel B, they increase up to 6 hours then decrease relative to time with Rad50-C47. Alternatively, if you compare it to wildtype, wildtype peaks around 4 hours and then decreases. Please be clear as to what is being referred to in the text on page 8 end of first paragraph.
5. Figure 4E, the image for Mre11 Wildtype does not agree with the plotted data. In the microscopy image, there is no visible Mre11 foci while the graph shows there ~10%.
6. Figure 4F, the data for Mre11 wildtype does not agree with the image in Figure 4E. Perhaps the

authors selected the wrong Mre11 wildtype image to show in panel E?

7. Figure 4G, is the data for this graph from the same data in panel E? At 4 hours, there is no double positive Mre11 and Rad51 foci in wildtype despite the graph in panel G indicating that there is ~10% of double Mre11 and Rad51 positive cells. Perhaps the authors used the wrong representative microscopy panel in the merge or the data for this plot is located elsewhere?

8. Page 8 middle paragraph, the authors refer to Mre11 foci peak in the 'early stage of meiosis while Rad51 foci peaked late in wild type cells'. Early and late are ambiguous. Please be more specific relative to the data that is presented.

9. The text in the middle paragraph related to Figure 4H does not appear to agree with the data in the figure. The *exo1-D173A* mutation did not abolish the accumulation of Rad51 foci. Rather, it greatly attenuated the number of Rad51 foci.

10. Page 8, last paragraph. The authors state that the intermediates did not increase further at a later time point referring to Figure I. In the gel in Panel I, the amount of the HD1 fragment is low until 6 hours, then there is an increase (~2 fold) and at 10 hours there is a decrease to less than the 8 hour amount of HD1 for the Rad50-C47. Please change the text on page 8 to be more accurate to the data that is shown. If the authors were trying to compare the results to those in the wildtype, perhaps consider indicating that comparison.

11. Figure 5A, release is misspelled.

12. Figure 5A, it isn't clear from the text, legend or methods why the band on the gel is labeled as TdT. Perhaps this band is end-labeled Spo11-DNA that is radioactively labeled by TdT?

13. Page 10, beginning of the second paragraph, the authors state the cleavage patterns by MRX-Sae2 and Spo11 are slightly different and that MRX-Sae2 acts on closer access points compared to Spo11. I am confused as to what is being compared here. MRX-Sae2 nuclease activity comes after Spo11 has cut the DNA and is bound to it. Comparing how MRX-Sae2 cuts to Spo11 is two different things. Perhaps what the authors mean is to compare the difference between Mre11-Rad50S-Xrs2-Sae2 to Mre11-Rad50C47-Xrs2-Sae2? Please clarify what is meant in this section.

Reviewer #2 (Remarks to the Author):

MRX (Mre11, Rad50, Xrs2) complex is a critical component of the DNA end processing machinery during double-strand break (DSB) repair in the budding yeast. It employs a combination of endo- and exonuclease activities to initiate DNA end resection in a manner stimulated by phospho-Sae2. Using a clever screening regime, the authors have identified a mutant of *rad50* (*rad50-C47*) within the protein's C-terminal ATP binding domain that confers resistance to hydroxyurea. Using recombinant MRX (*Rad50* WT or *rad50-C47*), the authors show that the C47 mutant is specifically defective in MRX exonuclease activity, especially when tested with Sae2. Endonuclease activity remains unaffected. Genetic analyses have revealed recombination defects during meiosis, including delayed DSB repair and Rad51 foci retention. Finally, the authors examine the impact of the *rad50-C47* mutant on processing of ends bound by a protein block (Spo11) or various hairpin structures. While there are two main DNA products formed in WT cells, *rad50-C47* cells generate a series of products that are larger. These results suggest decoupling of MRX's endo- and exonuclease activities by the *rad50-C47* mutation.

This is an impactful study that sheds light on MRX-Sae2 function and is clearly of general interest.

Specific Comments:

1. In the abstract, the authors refer to their separation-of-function mutant as "a defect in MRX 3'-exonuclease activity," whereas elsewhere, the defect is mentioned in the context of the MRX-Sae2 ensemble. The latter is the appropriate way to define this as no defect was shown with MRX alone in the exonuclease assay shown in Fig. 3C. Does the mutant *rad50* protein interact with pSae2 still?

2. As the strongest phenotypes of the *rad50-C47* are seen within the context of meiotic recombination, the authors should also measure Dmc1 foci similar to the approach used for Rad51 in Fig. 4E.

3. In Fig. 4E, the authors show that Rad51 foci are formed but not resolved in *rad50-C47* cells and

that resection is required to form these foci. They also suggest that these Rad51 filaments are not able to appropriately perform the subsequent strand invasion step. The authors indicate Spo11 is likely still bound to DNA ends, so are they assuming that this interferes with DNA strand invasion? Can they measure Spo11 foci retention in this context?

4. Does the rad50-C47 mutation affect ATPase activity of Rad50?

5. It would be important to test if rad50-C47 mutant is able to release other blocks such as that imposed by Ku.

6. Why do authors see more undigested substrate with EXO1+MRX+pSae2 combination compared to no EXO1? Have authors tested preincubation of MRX+Sae2 before addition of DNA substrate as in Fig. 3C?

Minor Comments:

a. Fig. 1E has dashed lines that do not appear to be referenced or identified.

b. Fig. 5A has a typo (release, not releae). What do the numbers on top of the panels designate (presumably time)?

c. Fig. S3 says Sae instead of Sae2.

Reviewer #3 (Remarks to the Author):

This manuscript by Tamai, Sugimoto, and colleagues describes the identification and characterization of two mutations in the Rad50 protein, a factor critical for double-strand break repair and signaling, using budding yeast as a model system. Mre11, a nuclease associated with Rad50, as well as the Xrs2 protein (MRX complex) are regulated by Sae2 to catalyze endonucleolytic and exonucleolytic processing of DNA ends. Previous work identified mutations in the N-terminus of Rad50 that abrogate Sae2 regulation of MRX (Rad50S mutants); here the C-terminal section of Rad50 is mutagenized and two mutants are found that affect MRX activities. The C-47 and C-126 mutants upregulate Tel1 activity (observed in a *mec1* background), a phenotype correlated with reduced end processing, previously identified in *sae2* mutants in yeast. In vivo, the mutations delay meiotic progression and the C-47 mutant in particular exhibits a marked delay in formation of meiotic DSB processing intermediates although is not as complete a block as Rad50S. In vitro analysis of the activity of the mutants shows that C-47 reduces the exonuclease activity of the complex while C-126 appears to affect both endo and exo activities. These results show that the exo activity of MRX is important for DNA end processing of some lesions, with meiotic Spo11-bound conjugates being most strongly affected.

Overall I think the most interesting aspect of the results shown is the demonstration of a role for the C-terminal conserved domain of Rad50 in regulation of Mre11 exonuclease activity. We still do not fully understand the relationship between Mre11, Rad50, and Xrs2 with respect to regulation of Mre11 nuclease activity and this work provides evidence for a more complex regulation of the exo activity than previously appreciated. That said, it is not very clear what the mechanistic basis of this phenomenon is since we do not have structural information about budding yeast MRX in DNA-bound conformations. Biologically, the results are very similar to a previously described exonuclease mutant of Mre11, H59S (also H59A mutants described), which was used by Neale and colleagues to first demonstrate the endo/exo model for meiotic recombination. This exo-deficient mutant was also shown to have essentially wild-type camptothecin resistance by Maizels and colleagues previously, similar to the work here. Endonuclease activity of Mre11 was also shown by Resnick and colleagues to be essential for hairpin opening. So the biological novelty of this exo-deficient Rad50 mutant (C-47) is somewhat limited, outside of the structural implications.

Other points:

1. What are the relative levels of the Rad50 mutant proteins in vegetatively growing cells and during meiosis? Are they comparable to the wild-type protein?

2. The Fig. 2C: rad50-C47 mutant shows lower survival than wild-type in this HO endonuclease-dependent assay for NHEJ. The conclusion here is that "the rad50-C47 mutation causes a more limited defect in MRX-Sae2 function compared with the rad50S mutation." (pg. 6). Since MRX actively participates in NHEJ in yeast though, is it not possible that this mutation negatively affects interactions with the NHEJ machinery? Are the associations of either C-47 or C-126 MRX complexes with Dnl4, Lif1, or Nej1 affected?

3. Fig. S3: The C126 mutant appears to have very similar endonuclease activity compared to WT Rad50, yet in the Rad53 phosphorylation assay (Fig. 1D), it looks similar to the sae2 deletion strain. Why is this and is it possible that this mutant has other defects? Does this mutant hyperstimulate Tel1 in a MEC1 background?
4. Fig. 5: The C47 Rad50 mutant is similar to the previously characterized H59S Mre11 mutant (Garcia et al, 2011) that is mostly deficient in exonuclease activity. The increase in size of Spo11-oligo fragments observed in this figure as well as the dependence of resection intermediates on Exo1 (Fig. 4) is also very similar to the H59S story. In light of this, I don't think the statement on pg. 10 is accurate: "Together, these studies using meiotic cells provide the first in vivo evidence that coordinated MRX-Sae2 endonuclease and 3'-5' exonuclease activities are required for the processing of blocked DNA double-strand ends".
5. Fig. 5A: Spo11 "release" misspelled; also position of resected strands in the diagram is incorrect.

Response to REVIEWER COMMENTS

We would like to thank the reviewers for their comments, which were extremely helpful in improving the manuscript. We revised the manuscript following their comments.

In the revision we showed that the *rad50-C47* mutation does not affect the expression level, MRX complex formation or MRX-Sae2 interaction. We also showed that the *rad50-C47* mutation does not affect ATP binding or hydrolysis even though the mutation is located near the ATP binding domain. We clearly explained that the *rad50-C47* mutation impairs Sae2-dependent activation of MRX 3'-5' exonuclease. Finally, we clarified ambiguities in the explanations of some experimental results. Changes in the revised manuscript were highlighted in red.

Responses to reviewers' general and specific comments are described below.

Reviewer #1

The manuscript entitled Sae2 controls Mre11 endo- and exonuclease activities by different mechanisms authored by Tomoki Tamai et al. The authors provide data that convincingly support the title of the manuscript regarding different mechanism of control by Sae2 of the endo and exonuclease activities of Mre11. The authors identified a separation of function rad50 mutant that is specifically defective in the RMX exonuclease activity. This allowed the author to show that both exo and endonuclease activity is needed to remove Spo11 from the ends of DNA. Contrastingly, the ability to remove hairpins by RMX relies on the endonuclease activity of Mre11. The author show that Sae2 promotes RMX cleavage at distinct distances from the Spo11-DNA end.

The results presented in this manuscript are significant and provide valuable insight to the regulation of RMX by Sae2. The work answers some unresolved questions in how RMX removed Spo11 from the ends of DNA. The work will be of great interest to the DNA repair field, NHEJ, and meiosis. The work supports the claims of the authors. Some of the text was a little unclear and could be revised to improve clarity for a broad audience. The authors interpretation of the data is logical. The methodology is sound. Some of the quantitation of the images is not in agreement with the images that are presented. This is likely a minor issue as the experiments were repeated multiple times and could be resolved by using images that are more representative of the data in the graphs. The quality of the work exceeds those expected in our field. The purified proteins and the in vitro digestion gels are beautiful. The authors have a number of methods that reference previous work. The description of the methods would allow someone in their field to replicate the experiments. Although my enthusiasm for this manuscript is relatively high, there are some concerns and some inconsistencies between the text and the data. I encourage the authors to address the concerns listed below as they will help with clarity of work.

Comments:

1. The authors performed structural analysis using UCSF Chimera and concluded that neither C47 and C126 mutations cause significant alterations. There was no mention of what structural analysis was performed. For instance, AlphaFold may not model the nuanced changes that could occur with a R to W or a K to R substitution. There are plenty of examples in the literature that involve a single amino acid change that results in changes in the structure of a protein that results in unexpected changes in the proteins behavior. While unlikely as the authors conclude, have the authors considered that the mutations may alter the affinity of Rad50 for Mre11-Xrs2 complex or the interaction of RMX with Sae2? Do the authors have results demonstrating that these mutants of RAD50 have the same affinity for

M-X and in the complex of RMX interacting with Sae2. Likewise, is it known whether the C47 and C126 mutations alter ATP affinity, ATP hydrolysis rates or DNA binding activity of RAD50. Given the proximity of the C47 and C126 mutations to the ATP binding pocket per Figure 1, it is possible that perturbation of any of these activities could provide clues to the mechanism by which Rad50 influences Mre11 activities with Sae2 present.

---- As suggested, we performed a more detailed biochemical characterization of the Rad50 mutants. We used mass photometry, a technique that relies on light scattering on a glass coverslip surface to measure the molecular weight of macromolecules to monitor the ability of the Rad50 variants to form the MRX complex together with MX. We show that wild-type Rad50 (Rad50-WT) and Rad50-C47 protein form dimers by themselves and interact with Mre11-Xrs2 (MX) to form the MRX complex with a 2:2:1 stoichiometry. These data are included as new **Fig. S3B-D**. We also tested the effect of *rad50-C47* or *rad50-C126* mutation on Mre11-Rad50 interaction *in vivo* by co-immunoprecipitation. Neither the *rad50-C47* nor *rad50-C126* mutation significantly affected Mre11-Rad50 complex formation (**Fig. 1H**).

The interaction with phosphorylated Sae2 (pSae2) was not stable enough to detect MRX-Sae2 complex formation in mass photometry. Instead, we monitored Rad50-pSae2 interaction by a pulldown assay using insect cells extract expressing MBP-pSae2 (new **Fig. 3G**). Rad50-C47 interacts with pSae2 to a similar degree as Rad50-WT. The above results are consistent with the idea that neither the *rad50-C47* nor the *rad50-C126* mutation leads to significant structural changes in the Rad50 protein.

The reviewer pointed out the possibility that the mutations could affect ATP binding, hydrolysis and ATP-dependent DNA binding. ATP binding of Rad50 inhibits MRX exonuclease activity (PMID: 30819891). We first examined the effect of the *rad50-C47* mutation on MRX exonuclease activity. Like MRX consisting of Rad50-WT, MRX consisting of Rad50-C47 exhibited decreased exonuclease activity in the presence of ATP or ATP- γ -S (new **Fig. 3H**). Moreover, Rad50-C47 was comparable to Rad50-WT in ATP-dependent DNA binding (new **Fig. 3I**). These results indicate that Rad50-C47 protein is proficient in ATP-binding.

The Rad50 ATPase activity is rather weak but can be stimulated by Rif2 (PMID: 26901759, PMID: 31640985). We examined the effect of the *rad50-C47* mutation on ATPase activity in the presence of Rif2. Our data indicate that Rad50-C47 protein is functionally equivalent to Rad50-WT in ATP hydrolysis (new **Fig. 3J**).

There was no mention of what structural analysis was performed.

---- In the revision we described that potential clashes of substituted amino acid residues were analyzed using the UCSF Chimera Rotamer tool in the method section. No clashes were detected with the ATP-binding form of Rad50, suggesting that no significant structural change occurs as the consequence. However, it remains possible that the structural changes are more evident in other forms during endo- or 3'-5' exonucleolytic cleavage (page 14-15, line 347-349).

2. Page 7 and Figure 3. The authors clearly show there are two different mechanisms by which Sae2 promotes endonuclease and exonuclease activities of Mre11 via Rad50. The purified proteins are exceptionally pure and the nuclease digestion gels are beautiful.

---- The reviewer did not suggest any modification.

3. In Figure 4B,C,D, the authors indicate that the signal for DNA fragments containing DSBI and DSBI decrease over time for the rad50-C47'. However, in the gel in Figure 4B, the DSB I and II is increasing over time for the rad50-C47 until a slight decrease at 10 hours. The data plotted in C for rad50-C47 do not reflect the band intensity in panel B. The authors may have selected the wrong representative gel to use in panel B?

----- The values in **Fig. 4C** are the ratio (%) of the combined signal intensity of DSB I and DSB II to the combined signal intensity of Total, including Parental (Pa), DSB I, and DSB II in **Fig. 4B**. Thus, the quantitative data in **Fig. 4C** correspond to kinetics as shown in **Fig. 4B**. The DSB I and II signal for *rad50-C47* cells decreased after 6 h (**Fig. 4C**) whereas the band intensity of DSB I or II did not (**Fig. 4B**). Correspondingly, the Pa band intensity increases after 6 h compared to 3 or 5 h (**Fig. 4C**). Thus, there is no error in presenting the data.

4. Figure 4D and B, the authors state that the faster migrating smears increased with *rad50-C47*. Please clarify what they were compared to and what they increased from? According to the gel in panel B, they increase up to 6 hours then decrease relative to time with *Rad50-C47*. Alternatively, if you compare it to wildtype, wildtype peaks around 4 hours and then decreases. Please be clear as to what is being referred to in the text on page 8 end of first paragraph.

----- In **Fig. 4D**, we used 4 h samples for quantifications as described in the legend. We also described that fast-migrating smeared signals were more pronounced in *rad50-C47* cells than in *rad50S* cells after 4 h of meiotic induction (**Fig. 4B and 4D**) (page 9, line 205-207).

5. Figure 4E, the image for *Mre11 Wildtype* does not agree with the plotted data. In the microscopy image, there is no visible *Mre11* foci while the graph shows there ~10%.

----- The plot indicates that 10% of wild-type cells contained at least five *Mre11* foci at 4 h whereas 90% of wild-type cells had less than five foci. Therefore, the representative microscopic image for wild-type cells do not have visible foci.

6. Figure 4F, the data for *Mre11 wildtype* does not agree with the image in Figure 4E. Perhaps the authors selected the wrong *Mre11 wildtype* image to show in panel E?

----- The experiments in **Fig. 4E and 4F** are different but complementary. The plot in **Fig. 4F** (also **4H**) shows the number of Rad51 or Mre11 foci within foci positive cells. Foci positive cells are defined as those containing more than five foci. In contrast, the plot in **Fig. 4E** shows the percentage of foci positive cells.

7. Figure 4G, is the data for this graph from the same data in panel E? At 4 hours, there is no double positive *Mre11* and *Rad51* foci in wildtype despite the graph in panel G indicating that there is ~10% of double *Mre11* and *Rad51* positive cells. Perhaps the authors used the wrong representative microscopy panel in the merge or the data for this plot is located elsewhere?

----- The plot in **Fig. 4G** shows that 10% of wild-type cells are both *Mre11*- and *Rad51*-foci positive at 4 h. In other words, most wild-type cells (90%) are not double-positive (**Fig. 4G**). The microscopic image in **Fig. 4E** shows a representing image that cells are not double-positive (*Rad51*-positive but *Mre11*-negative).

8. Page 8 middle paragraph, the authors refer to *Mre11* foci peak in the 'early stage of meiosis while *Rad51* foci peaked late in wild type cells'. Early and late are ambiguous. Please be more specific relative to the data that is presented.

----- We included time points (hour) after meiosis induction, as suggested (see page 9, line 213-214).

9. The text in the middle paragraph related to Figure 4H does not appear to agree with the data in the figure. The *exo1-D173A* mutation did not abolish the accumulation of *Rad51* foci. Rather, it greatly attenuated the number of *Rad51* foci.

----- We modified the sentence as suggested (see page 10, line 219)

10. Page 8, last paragraph. The authors state that the intermediates did not increase further at a later time point referring to Figure 4 I. In the gel in Panel I, the amount of the HD1 fragment is low until 6 hours, then there is an increase (~2 fold) and at 10 hours there is a

decrease to less than the 8 hour amount of HD1 for the Rad50-C47. Please change the text on page 8 to be more accurate to the data that is shown. If the authors were trying to compare the results to those in the wildtype, perhaps consider indicating that comparison. ----- As suggested by the reviewer, we modified the paragraph and indicated the time points (see page 10, line 229-231).

We note that the HD1 signals in *rad50S* and *rad50-C47* cells were overlapped with non-specific signals (see the figure legend of **Fig. 4I**); only the HD2 and HD3+4 signals were valid for evaluation.

11. Figure 5A, release is misspelled.

----- We corrected the typo.

12. Figure 5A, it isn't clear from the text, legend or methods why the band on the gel is labeled as TdT. Perhaps this band is end-labeled Spo11-DNA that is radioactively labeled by TdT?

----- We added the following sentence in the figure legend of **Fig. 5A**. "The asterisk indicates non-specific bands derived from oligonucleotides contaminated in commercially available TdT (PMID:16107854)."

13. Page 10, beginning of the second paragraph, the authors state the cleavage patterns by MRX-Sae2 and Spo11 are slightly different and that MRX-Sae2 acts on closer access points compared to Spo11. I am confused as to what is being compared here. MRX-Sae2 nuclease activity comes after Spo11 has cut the DNA and is bound to it. Comparing how MRX-Sae2 cuts to Spo11 is two different things. Perhaps what the authors mean is to compare the difference between Mre11-Rad50S-Xrs2-Sae2 to Mre11-Rad50C47-Xrs2-Sae2? Please clarify what is meant in this section.

----- We modified the paragraph as follows.

Interestingly, similar but more robust ladder bands were observed in *rad50-C47* mutant cells compared with *rad50S* mutant cells, indicating that MRX-Sae2 also accesses DNA every 10 nt near DNA-bound Spo11 (see page 11, line 260).

Reviewer #2

MRX (*Mre11*, *Rad50*, *Xrs2*) complex is a critical component of the DNA end processing machinery during double-strand break (DSB) repair in the budding yeast. It employs a combination of endo- and exonuclease activities to initiate DNA end resection in a manner stimulated by phospho-Sae2. Using a clever screening regime, the authors have identified a mutant of *rad50* (*rad50-C47*) within the protein's C-terminal ATP binding domain that confers resistance to hydroxyurea. Using recombinant MRX (*Rad50* WT or *rad50-C47*), the authors show that the C47 mutant is specifically defective in MRX exonuclease activity, especially when tested with Sae2. Endonuclease activity remains unaffected. Genetic analyses have revealed recombination defects during meiosis, including delayed DSB repair and *Rad51* foci retention. Finally, the authors examine the impact of the *rad50-C47* mutant on processing of ends bound by a protein block (*Spo11*) or various hairpin structures. While there are two main DNA products formed in WT cells, *rad50-C47* cells generate a series of products that are larger. These results suggest decoupling of MRX's endo- and exonuclease activities by the *rad50-C47* mutation.

This is an impactful study that sheds light on MRX-Sae2 function and is clearly of general interest.

Specific Comments:

1. In the abstract, the authors refer to their separation-of-function mutant as "a defect in MRX

3'-exonuclease activity," whereas elsewhere, the defect is mentioned in the context of the MRX-Sae2 ensemble. The latter is the appropriate way to define this as no defect was shown with MRX alone in the exonuclease assay shown in Fig. 3C.

----- We corrected the sentence in the context of the MRX-Sae2 ensemble (Abstract, page 2, line 7) and modified the other sentences in the main text to make this point clear.

Does the mutant rad50 protein interact with pSae2 still?

----- We performed pulldown assays and found that the Rad50-C47 mutant interacted with pSae2 similarly to wild-type Rad50. The result was shown in a new figure (see **Fig. 3G**). This finding is consistent with the observation that pSae2 stimulated the endonuclease activity of MRX consisting of Rad50-C47.

2. *As the strongest phenotypes of the rad50-C47 are seen within the context of meiotic recombination, the authors should also measure Dmc1 foci similar to the approach used for Rad51 in Fig. 4E.*

----- Dmc1, a meiosis-specific Rad51 homolog, is essential for meiotic recombination. Like Rad51, Dmc1 binds to the 3'-ssDNA overhangs and form discrete foci. One explanation for strong meiotic recombination phenotypes observed in *rad50-C47* mutants is the lack of Dmc1 accumulation near DSB ends. We monitored the formation of Dmc1 foci in *rad50-C47* mutants. Dmc1 formed foci in *rad50-C47* cells (**Fig. S4**). These results further support that ssDNA is generated after endonucleolytic cleavage in *rad50-C47* cells during meiosis. Dmc1 foci do not fully overlap with Rad51 foci, suggesting Dmc1 and Rad51 have different ssDNA binding properties (PMID:26719980, PMID:32610038). One remaining possibility is that Dmc1 and Rad51 localize imprecisely to ssDNA filaments in *rad50-C47* mutants, thus resulting in meiotic recombination defect. We described the possibility that uncoupled ssDNA binding of Rad51 and Dmc1 affects strand invasion or exchange in *rad50-C47* mutants during meiotic recombination (see page 17, line 401-402).

3. *In Fig. 4E, the authors show that Rad51 foci are formed but not resolved in rad50-C47 cells and that resection is required to form these foci. They also suggest that these Rad51 filaments are not able to appropriately perform the subsequent strand invasion step. The authors indicate Spo11 is likely still bound to DNA ends, so are they assuming that this interferes with DNA strand invasion? Can they measure Spo11 foci retention in this context?*

----- Spo11 transiently associates with meiotic DSB hotspots independently of DSB induction (PMID:15655113). Thus, Spo11 foci formation does not necessarily indicate that Spo11 is covalently attached to DNA ends.

The experiments using denatured gel-electrophoresis (**Fig. 5**) indicate that Spo11 coprecipitates with short DNA oligonucleotides longer than 30 nt without the addition of any cross-linking reagents. Spo11 is covalently bound to 5'-end of DNA of the cleaved strand and MRX-Sae2 endonuclease introduces nicks specifically in the 5'-strand. Thus, our results support the idea that Spo11 is covalently retained to the DNA end in *rad50-C47* mutants.

While ssDNA was generated, meiotic recombination was nearly blocked in *rad50-C47* mutants. One explanation is that Spo11 retained at the DNA end or a short unresected double-stranded DNA region near Spo11 prevents strand invasion or exchange in *rad50-C47* mutants during meiosis. The observation that hairpin formation interferes with homologous recombination supports this view. However, it remains possible that uncoupled ssDNA binding of Rad51 and Dmc1 affects strand invasion or exchange in *rad50-C47* mutants during meiosis. We discussed these points in the Discussion (see the reviewer's specific comment 2).

4. *Does the rad50-C47 mutation affect ATPase activity of Rad50?*

----- We found that the *rad50-C47* mutation does not affect the ATPase activity of Rad50 *in vitro* (**Fig. 3J**).

5. It would be important to test if rad50-C47 mutant is able to release other blocks such as that imposed by Ku.

----- As suggested by the reviewer, we performed nuclease assays using Ku-bound substrates. The Rad50-C47 variant was slightly defective in nicking next to Ku compared to Rad50-WT, while the Rad50-C126 variant was significantly more compromised than Rad50-WT. These new results were shown in Fig. S3A.

6. Why do authors see more undigested substrate with EXO1+MRX+pSae2 combination compared to no EXO1? Have authors tested preincubation of MRX+Sae2 before addition of DNA substrate as in Fig. 3C?

----- The reviewer wondered whether the lower substrate utilization in the presence of Exo1 is due to non-productive Exo1 binding to the DNA substrate. For example, a small fraction of Exo1 in a protein preparation may be catalytically inactive, thereby competing with catalytically active Exo1 for substrate.

In support of this explanation, if we preincubate the substrate with MRX-pSae2 (5 min on ice) before the addition of Exo1 and lower the concentration of Exo1 (from 25 nM to 5 nM), the difference in substrate utilization is lower (See Figure below, condition B). We note, however, that the products generated by MRX-pSae2 are not degraded as efficiently, probably due to the lower Exo1 concentration. Hence, we opted for leaving the original Figure in the manuscript (condition A). Furthermore, we modified the text to clarify that Exo1 extends the products of MRX-pSae2 endonuclease activity.

Minor Comments:

a. Fig. 1E has dashed lines that do not appear to be referenced or identified.

----- We added the explanation for it in the figure legend.

b. Fig. 5A has a typo (release, not releae). What do the numbers on top of the panels designate (presumably time)?

----- We corrected the typo and added required information.

c. Fig. S3 says Sae instead of Sae2.

----- We corrected the typo.

Reviewer #3 (Remarks to the Author):

This manuscript by Tamai, Sugimoto, and colleagues describes the identification and characterization of two mutations in the Rad50 protein, a factor critical for double-strand break repair and signaling, using budding yeast as a model system. Mre11, a nuclease associated with Rad50, as well as the Xrs2 protein (MRX complex) are regulated by Sae2 to catalyze endonucleolytic and exonucleolytic processing of DNA ends. Previous work identified mutations in the N-terminus of Rad50 that abrogate Sae2 regulation of MRX (Rad50S mutants); here the C-terminal section of Rad50 is mutagenized and two mutants are found that affect MRX activities. The C-47 and C-126 mutants upregulate Tel1 activity (observed in a mec1 background), a phenotype correlated with reduced end processing, previously identified in sae2 mutants in yeast. In vivo, the mutations delay meiotic progression and the C-47 mutant in particular exhibits a marked delay in formation of meiotic DSB processing intermediates although is not as complete a block as Rad50S. In vitro analysis of the activity of the mutants shows that C-47 reduces the exonuclease activity of the complex while C-126 appears to affect both endo and exo activities. These results show that the exo activity of MRX is important for DNA end processing of some lesions, with meiotic Spo11-bound conjugates being most strongly affected.

Overall I think the most interesting aspect of the results shown is the demonstration of a role for the C-terminal conserved domain of Rad50 in regulation of Mre11 exonuclease activity. We still do not fully understand the relationship between Mre11, Rad50, and Xrs2 with respect to regulation of Mre11 nuclease activity and this work provides evidence for a more complex regulation of the exo activity than previously appreciated.

That said, it is not very clear what the mechanistic basis of this phenomenon is since we do not have structural information about budding yeast MRX in DNA-bound conformations. Biologically, the results are very similar to a previously described exonuclease mutant of Mre11, H59S (also H59A mutants described), which was used by Neale and colleagues to first demonstrate the endo/exo model for meiotic recombination. This exo-deficient mutant was also shown to have essentially wild-type camptothecin resistance by Maizels and colleagues previously, similar to the work here.

----- In the revision, we pointed out that the *rad50-C47* mutation is a new type of *rad50* mutation, which is different from the *mre11-H59S* mutation in several aspects. The *mre11-H59S* mutation affected the basal Mre11 exonuclease activity independently of Sae2 *in vitro*. However, the *rad50-C47* mutation impairs Sae2-dependent activation of MRX exonuclease activity but does not the basal activity *in vitro*. In this revision, we clearly described this point.

In vivo, cells carrying the *mre11-H59S* single mutation undergo meiosis (Neale and colleagues, PMID:22002605). In contrast, cells carrying the *rad50-C47* mutation did not progress meiosis as found for the *rad50S* mutation (See below our response to other points 4 of the reviewer's comments). Finally, we would like to point out that the studies by Resnick and colleagues (PMID:1182209) did not discriminate the MRX-Sae2 endonuclease activity from the 3'-5' exonuclease activity in hairpin opening.

So the biological novelty of this exo-deficient Rad50 mutant (C-47) is somewhat limited, outside of the structural implications.

----- Recent structural analysis indicates that MR recognizes similarly structured DNA substrates and triggers the action of endonucleases and 3'-5' exonucleases. Our results show that Sae2 stimulates the MRX endonuclease and the 3'-5' exonuclease by different mechanisms, suggesting that MRX-Sae2 undergoes distinct structural changes when acting as an endonuclease or 3'-5' exonuclease. As pointed out by the reviewer, there is no

structural information for MRX in DNA-bound conformations. Our results should accelerate the structural analysis of MRX in DNA end processing.

Other points:

1. *What are the relative levels of the Rad50 mutant proteins in vegetatively growing cells and during meiosis? Are they comparable to the wild-type protein?*

---- In this revision we showed that neither the *rad50-C47* nor the *rad50-C126* mutation significantly affected the expression levels of Rad50 during vegetative growth conditions and meiosis (see **Fig. 1G** and **Fig. 2B**).

2. *The Fig. 2C: rad50-C47 mutant shows lower survival than wild-type in this HO endonuclease-dependent assay for NHEJ. The conclusion here is that "the rad50-C47 mutation causes a more limited defect in MRX-Sae2 function compared with the rad50S mutation." (pg. 6). Since MRX actively participates in NHEJ in yeast though, is it not possible that this mutation negatively affects interactions with the NHEJ machinery? Are the associations of either C-47 or C-126 MRX complexes with Dnl4, Lif1, or Nej1 affected?*

---- Our data show that the *rad50-C47* mutation causes defects only in the 3'-5' endonuclease, whereas the *rad50-C126* mutation causes defects in both the endonuclease and the 3-5 exonuclease. Our data are consistent with the current model in which DNA end resection attenuates NHEJ and explain why the *rad50-C126* mutation enhances NHEJ similarly to *rad50S* or *sae2Δ* mutation whereas the *rad50-C47* mutation does not. We noted that cells carrying the *rad50-C47* single mutation had slightly reduced NHEJ efficiency compared to wild-type cells. However, introduction of the *rad50-C47* mutation did not cause obvious NHEJ defects in the *sae2Δ* mutant; the survival rate of *sae2Δ* single and *sae2Δ rad50-C47* double mutants are very similar. Thus, the significance of the weak NHEJ defect observed in *rad50-C47* single mutants is not clear (page 7, line 141-144). Our data do not fully exclude the possibility that the *rad50-C47* or the *rad50-C126* mutation negatively affects the interaction of MRX with NHEJ factors, but the impact is likely to be minor.

3. *Fig. S3: The C126 mutant appears to have very similar endonuclease activity compared to WT Rad50, yet in the Rad53 phosphorylation assay (Fig. 1D), it looks similar to the sae2 deletion strain. Why is this and is it possible that this mutant has other defects? Does this mutant hyperstimulate Tel1 in a MEC1 background?*

---- Several studies have already shown that Mre11 nuclease activity does not affect the Tel1 signaling pathway. The current model (Symington and Longhese groups, PMID:30510002, PMID: 30538107) suggests that Sae2 inhibits Rad9-Rad53 interaction near DNA damage sites, thereby attenuating Rad53 phosphorylation. Note that Rad9 is a checkpoint mediator that connects Mec1 and Tel1 to Rad53. Both the *rad50-C47* and *rad50-C126* mutations carry substitution mutations at the C-terminus. In the revision we extended the discussion and pointed out that Sae2 may control Rad9-Rad53 interaction through the C-terminus of Rad50 (see Discussion, page 18, line 435-438).

4. *Fig. 5: The C47 Rad50 mutant is similar to the previously characterized H59S Mre11 mutant (Garcia et al, 2011) that is mostly deficient in exonuclease activity. The increase in size of Spo11-oligo fragments observed in this figure as well as the dependence of resection intermediates on Exo1 (Fig. 4) is also very similar to the H59S story. In light of this, I don't think the statement on pg. 10 is accurate: "Together, these studies using meiotic cells provide the first in vivo evidence that coordinated MRX-Sae2 endonuclease and 3'-5' exonuclease activities are required for the processing of blocked DNA double-strand ends".*

---- Previous studies have shown that *mre11-H59S* single mutants are not significantly defective in meiotic DSB processing; only *mre11-H59S exo1* double mutants showed significant defects (see fig. 1b of Garcia et al, 2011, PMID:22002605). Thus, it was not clear whether the 3'-5' exonuclease of MRX-Sae2 is critical for the processing of meiotic DSB breaks. In this study we show that the *rad50-C47* mutation by itself causes a defect in meiotic DSB processing (**Fig. 4C**, in this study). We discussed this point in the revision (See

the Discussion, page 16, line 378-385). Thus, our studies provide the first evidence that, in addition to the endonuclease activity, the 3'-5' exonuclease activity of MRX-Sae2 is critical for the processing of meiotic DSB breaks.

5. Fig. 5A: Spo11 "release" misspelled; also position of resected strands in the diagram is incorrect.

----- We corrected the errors.

REVIEWERS' COMMENTS

Reviewer #2 (Remarks to the Author):

Authors have done an admirable job addressing my concerns.

This is an impactful study that sheds light on how MRX-Sae2 facilitates processing of meiotic DNA breaks to promote homologous recombination and repair.

Reviewer #3 (Remarks to the Author):

The authors have addressed all of my previous concerns.

Response to REVIEWER COMMENTS

The reviewers did not point out any issues with the current version.